# Inundation and evacuation of shoreline populations during landslide-triggered tsunami: An integrated numerical and statistical hazard assessment

Emmie Malika Bonilauri[1], Catherine Aaron[2], Matteo Cerminara[3], Raphaël Paris[1], Tomaso Esposti Ongaro[3], Benedetta Calusi[3,4], Domenico Mangione[5] and Andrew John Lang Harris[1]

[1]Laboratoire Magmas et Volcans, OPGC, IRD, CNRS, Université Clermont Auvergne, F- 63000 Clermont – Ferrand, France
[2]Laboratoire de Mathématiques Blaise Pascal, CNRS, Université Clermont Auvergne, Clermont – Ferrand, 63170, France
[3]Istituto Nazionale di Geofisica e Vulcanologia (INGV), Sezione di Pisa, Pisa, 56125, Italy
[4]Dipartimento di Matematica e Informatica Ulisse Dini, Università degli Studi di Firenze, Firenze, 50134, Italy
[5]Dipartimento della Protezione Civile, Rome, 00189, Italy

*Correspondence to*: Emmie M. Bonilauri (Emmie.BONILAURI@uca.fr, emmie.bonilauri@gmail.com)

**Abstract.** The volcanic island of Stromboli (Southern Tyrrhenian sea, Italy) is renowned for its persistent, periodic, low-intensity explosive activity, whose spectacular manifestations attract tens of thousands of tourists every year. However, sporadic more intense major explosive and effusive eruptions, and paroxysms, pose serious threats to the island. In addition to direct hazards, granular slides of volcanic debris and pyroclastic avalanches, which can rapidly reach the sea potentially generating tsunamis, are often associated with such unpredictable eruptive activity. Due to the very fast propagation of the tsunami around the island, and the consequent short tsunami warning time (ranging from less than a minute to only a few minutes) mitigation efforts and evacuation from the Strombolian coast must be carefully planned. In this paper, we describe a new, GIS-assisted procedure that allows us to combine the outputs of an ensemble of 156 pre-computed landslide-generated tsunami hazard scenarios (with variable landslide volume, position, and density), statistical exposure data (i.e., the number of inhabitants and tourists) and digital geographic information, to obtain a quantitative (scenario-based) risk analysis. By means of the analysis of the road network and coastal morphology, we develop a model with routes and times to reach a safe area from every pixel in the inundated area, and appraisal for the time needed to escape versus the wave arrival time. This allows us to evaluate and quantify the effectiveness of potential risk mitigation by means of evacuation. The creation of an impact score linking the predicted inundation extent and the tsunami warning signals is intended, in the long term, to predict the intensity of future tsunamis, and to adapt evacuation plans accordingly. The model, here applied to Stromboli, is general, and can be applied to other volcanic islands. Evacuating an island hosting several thousand tourists every summer with very little warning time supports the absolute necessity for such mitigation efforts, aimed at informing hazard planners and managers, and all other stakeholders.

## 1 Introduction

The NOAA's NCEI (National Center for Environmental Information) estimates that about 81% of recorded tsunamis originate from earthquakes, with 7% originating from landslides and 5% from volcanoes, the remaining 5% being from an unknown source (NCEI; Harbitz et al., 2014). Landslide-generated tsunamis vary greatly in their size and origin, with volcano flank collapse being a frequent source. The range of volume and position of the collapse on the volcano flank produces great variation in terms of coastal impact (Grezio et al., 2017). Large volume collapse of flank of a volcano island can, for example, generate local tsunamis with waves tens of metres high in the proximal fields (Paris et al., 2018). However, tsunamis caused by volcanic landslides are characterized by shorter wavelengths compared to earthquake-induced tsunamis (Grezio et al., 2017), and consequently their far-field effect is often considerably reduced compared to their proximal impact as observed, e.g., during the 2018 Anak Krakatau tsunami (Muhari et al., 2019). However, even a 1 m high tsunami inundating a beach can present a high risk to populated coastlines and tourist seaside resorts.

At Stromboli, an active volcano of the Aeolian archipelago (Southern Tyrrhenian sea, Italy; Rosi et al., 2000, 2013) tsunamis can be potentially generated by instability and collapse of volcano flanks (in particular, the steep north-western slope of the Sciara del Fuoco) or by pyroclastic currents generated by paroxysmal eruptions (Pistolesi et al., 2020; Giordano and De Astis, 2020). One example of a tsunami generated by a landslide is that of Stromboli in December 2002. In that case, two landslides generated two tsunamis 7 minutes apart with maximum run-up of 10.9 m (Tinti et al., 2006a). The first tsunami was due to a near-shore submarine landslide, probably involving up to $20 \times 10^6$ m$^3$ of material (Chiocci et al., 2003), and the second by a subaerial landslide of $4 - 9 \times 10^6$ m$^3$ that broke off at about 500 m a.s.l (Tinti et al., 2006b). The resulting tsunami caused significant damage along the island coast up to an elevation of about 10 m above sea-level (Tinti et al., 2006a; Fornaciai et al., 2022), as well as at the island of Panarea 21 km distant (Maramai et al., 2005a).

The risk associated with tsunamis is particularly high on Stromboli, due the high density of population on the coasts, and where travel time for the tsunami from the source is minutes (Bonilauri et al., 2021). This is enhanced at Stromboli where, with the development of global tourism, vacation rentals, restaurants, shops, and hotels have been built close to the beaches. With the development of tourism and influx of seasonal workers, the collective memory linked to the tsunami risk also becomes reduced (Tulius, 2020; Riskianingrum and Yogaswara, 2022).

For these reasons, after the 2002 events, the scientific community and the Italian Civil Protection Department undertook a series of initiatives aimed at quantifying the hazards associated with landslide-generated tsunamis, providing an early detection and alert system, to mitigate the associated risks. In particular, the Laboratorio di Geofisica Sperimentale (LGS) of the University of Florence installed two tsunami detection beacons: the first in 2008 at the South-West of the foot of the Sciara del Fuoco and the second in 2017 at the North-East (Selva et al., 2021). To be able to generate an automatic alert and warn the relevant authorities and the population as quickly as possible, they have developed a tsunami detection algorithm by studying the STA/LTA (Short Time Average over Long Time Average) ratio and the dispersion of surface waves (Lacanna and Ripepe, 2020; Selva et al., 2021). They were able to test the algorithm successfully during the two paroxysms in July and August 2019

(Lacanna and Ripepe, 2020; Selva et al., 2021) and during the avalanche caused by the partial collapse of the NE crater on the 4th of December of 2022.

In parallel, a systematic study has been carried out to define tsunami hazard scenarios based on several, complementary approaches. The first is the identification of past events in the sedimentological record (tsunami deposits) and historical archives in order to build a tsunami catalogue (Maramai et al., 2014; Pistolesi et al., 2020). The second is based on the interpretation of observations and instrumental data (Bonaccorso et al., 2003; Pino et al., 2004; Boldini et al., 2005; Calvari et al., 2005; Maramai et al., 2005a,b; Tinti et al., 2005; Tommasi et al., 2005; Acocella et al., 2006; Tinti et al., 2006a,b; Chiocci

et al., 2008a,b; Fornaciai et al., 2022). The third comprises model-based scenarios through data-driven simulations (Fornaciai et al., 2019; Bonilauri et al., 2021; Esposti Ongaro et al., 2021, Cerminara et al., 2024).

Finally, a series of initiatives have been carried out to enhance the awareness about the tsunami risk and to enforce mitigation actions aimed at the timely evacuation of the risk zones in case of tsunami via the "Io non rischio" awareness campaign (https://www.iononrischio.gov.it/en/get-ready/volcanoes/stromboli/what-do/) and by several work tasks as the ongoing Task

4.3 of DPC-INGV 2022-2024 Agreement for the implementation of the service activities: "Survey on risk perception of explosive paroxysms and tsunamis to better define a communication strategy and informative materials".

In this paper, we describe a new procedure that allows us to combine the outputs of an ensemble of pre-computed tsunami hazard scenarios and exposure data and digital geographic information, to obtain a quantitative (scenario-based) risk analysis and to quantify the effectiveness of potential risk mitigation by means of evacuation. In our procedure, the tsunami hazard is

represented by a set of (static, georeferenced raster) maps of inundation (maximum wave height above ground) and wave arrival times, one for every individual scenario produced by numerical simulations. Every tsunami scenario is in turn defined, in a deterministic approach, by a corresponding landslide scenario, and it is associated with a trigger time (which approximately corresponds to the initial time at which an alert is issued). For every scenario, we are able to calculate an "impact score", which classifies in a simple and intuitive way the scenarios in terms of their impact on the island shores. For exposure, we consider

the population distribution on the island and the geometry of the road network, with which we can compute escape times from any pixel of the map towards a Refuge Area Entry Point (RAEP). By convolving hazard and exposure maps, we are then able to obtain maps expressing the level of risk of the different areas along the Stromboli shores, in terms of potential impact of a given scenario and potential mitigation in terms of evacuation capacity.

The procedure here applied to the island of Stromboli is a general one, and we discuss in the following sections its main

components, assumptions, and criticalities.

## 2 Methods

### 2.1 Tsunami hazard scenarios

In this paper, we base our analysis on tsunami scenarios produced by numerical simulations with a coupled landslide-water multilayer, non-hydrostatic shallow water model (Esposti Ongaro et al., 2021, Cerminara et al., 2024). The landslide is

considered as a granular fluid, having a given initial volume, position, and density, which is dynamically two-way coupled with the water layers. The Multilayer-HySEA (Hyperbolic Systems and Efficient Algorithms) numerical model (Fernández-Nieto et al., 2018; Macías et al., 2021a,b) adopted to generated the scenarios is particularly suited to the case of a volcanic island, since typical landslide-generated tsunamis have short wavelengths and develop over steep topo-bathymetry, making the usual hydrostatic approximation fail.

Simulations using Multilayer-HySEA were performed using ten different initial landslide positions (positions 1 to 10, Figure 1), five different volumes (5, 8, 14, 21, and $30 \times 10^6$ m$^3$) and three different densities (2.5, 2.0 and 1.7 kg/m$^3$, or water/landslide density contrasts of 0.4, 0.5, 0.6, respectively), basing on the ranges hypothesized for the 2002 tsunami event at Stromboli (Fornaciai et al., 2019; Esposti Ongaro et al., 2021; Cerminara et al., 2024). Additional simulations from a higher subaerial position (Position 0, Figure 1) were run with two different volumes, for a total of 156 simulations analysed. For each

simulation, we consider (1) the coastal inundation (i.e., the estimated inundation depth for each onshore pixel); (2) the tsunami arrival time for each pixel in the inundated area; (3) the offshore water surface elevation.

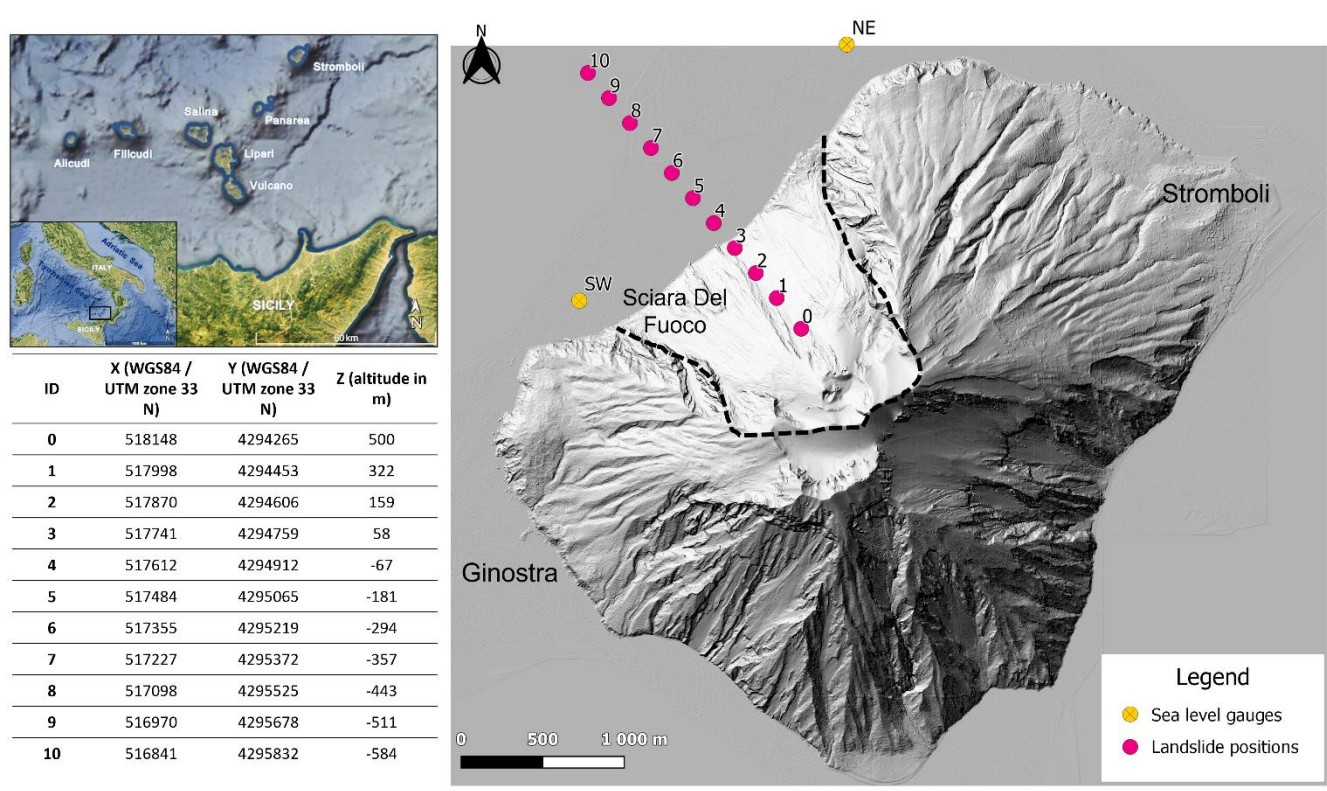

| ID | X (WGS84 / UTM zone 33 N) | Y (WGS84 / UTM zone 33 N) | Z (altitude in m) |
|---|---|---|---|
| 0 | 518148 | 4294265 | 500 |
| 1 | 517998 | 4294453 | 322 |
| 2 | 517870 | 4294606 | 159 |
| 3 | 517741 | 4294759 | 58 |
| 4 | 517612 | 4294912 | -67 |
| 5 | 517484 | 4295065 | -181 |
| 6 | 517355 | 4295219 | -294 |
| 7 | 517227 | 4295372 | -357 |
| 8 | 517098 | 4295525 | -443 |
| 9 | 516970 | 4295678 | -511 |
| 10 | 516841 | 4295832 | -584 |

**Figure 1: Location of Stromboli Island and landslide position numbers simulated by the INGV of Pisa with their respective coordinates and altitudes, position number 0 corresponding to December 2002 characteristics. The authors made maps by using**
**Quantum GIS version 3.16.7 software (2021).**

For each simulation, we also define an onset time $T_0$, when the landslide starts, a final time $T_{600}$, when the simulation is stopped, and "trigger time" $T_g$, when the generated wave overcomes a given threshold at one of the two gauges offshore the Sciara del

Fuoco. These are located on the position of the actual beacons, installed close to Punta Labronzo (North East Beacon, BNE) and Punta dei Corvi (SouthWest Beacon, BSW). In this work, we decided to adopt a threshold of +0.3 m or -0.3 m on the wave

detection at the gauges, to analyse only "significant" scenarios in terms of impact. The choice of such a threshold was done on a subjective basis and might be the object of further analysis in the future.

The trigger time $T_g$ should be approximatively equivalent to the time at which the alert is issued, following the procedure described by Selva et al (2021) and Lacanna and Ripepe (2020). Finally, the wave arrival time at a given pixel is computed as the difference between the actual tsunami arrival time and the trigger time $T_g$.

All simulation and post-processed data were integrated in QGIS (Quantum GIS) 3.16.7 with GRASS (Geographic Resources Analysis Support System) version 7.8.5. Topography was represented by a Digital Elevation Model (DEM) of 31635 pixels (20 m by 20 m) from the LiDAR (Light Detection And Ranging) campaign carried out in July 2010 by the INGV and bathymetry was from the Marine Geohazards along the Italian Coasts (MaGIC) project (for more details, see: Favalli et al., 2009; Chiocci and Ridente, 2011; Fornaciai et al., 2019).

**2.2 Inundation Impact Score and link with the proximal wave height**

For each landslide and tsunami scenario, we define an "impact score" S, equal to the number of on land pixels (in the digital model) that are inundated at a given time, during the numerical simulation (we might also consider our impact score in terms of inundated area in m²). The impact score allows us to classify the landslide-tsunami scenarios based on their coastal hazard and to link such a hazard to the features of the tsunami at the proximal gauge. In particular, we have used several statistical

methods to try to establish a robust link between the impact score and the maximum wave height at the proximal gauge, which is, in principle, a measurable quantity. The classic method for this type of problem (that is, to find an unknown value from several knowns) is linear regression. This approach, however, lacks robustness when the number of explanatory variables (here, 40) is too large compared with the number of individuals (here, the 156 simulations). LASSO linear regression is thus a modification of traditional linear regression that identifies a subset of explanatory variables (in this case, times of interest for

measuring wave heights) of sufficiently small size for the results to be robust.

In a real case, the volume, density or position of the landslide is not immediately known, thus we only have the signals of the two gauges. As a result, the aim of LASSO is to find out whether there is any chance of detecting the impact of tsunamis on coastlines both quickly and accurately (without counterproductive false alarms) before the tsunami arrives and without knowledge of the landslide characteristics. We thus determine how we can use simulated wave signals to correctly determine

the impact score. To set the impact score, we used the signals from the 2 beacons, i.e., 40 variables (1 wave height every 2 seconds between 0 and 40 seconds for 2 beacons). The statistical procedure is described in detail in Appendix A.

**2.3 Pedestrian horizontal evacuation model**

For every simulated scenario, we applied an evacuation model to each inundated pixel, by using the approach of Bonilauri et al. (2021). This is a macroscopic model, i.e., a model for global evacuation and not an agent-based approach, which determines

the fastest pedestrian evacuation paths from a danger point to a safe point, and compares the escape time with the wave arrival time.

To apply the evacuation model, a grid with the same mesh size as the inundation model (20 m) was created for the village of Stromboli (Figure 2), and a centroid, i.e., an evacuation starting point, was placed central to each pixel. A speed reduction coefficient was assigned to each road segment according to its width, slope, and surface type (Table 1). For each tsunami

simulation, the quickest escape routes were projected from each inundated centroid, and times required to evacuate each inundated pixel were calculated; with evacuation time being the time needed to move from the inundated pixel to a "safe" zone defined by the limit of the area impacted by the tsunami.

To reach the safe zone, evacuees must pass through a Refuge Area Entry Points (RAEP). Two categories of RAEP were established:

1.   A *normal-event* RAEP was assigned for tsunamis with run-ups less than or approximately equal to that of the December 2002 event, i.e., the RAEP was placed at the intersection between road network and 15 m contour line.

    2.   An *extreme-event* RAEP was set for tsunamis greater than the December 2002 run-up, and placed at the intersection between road network and 35–40 m contour line.

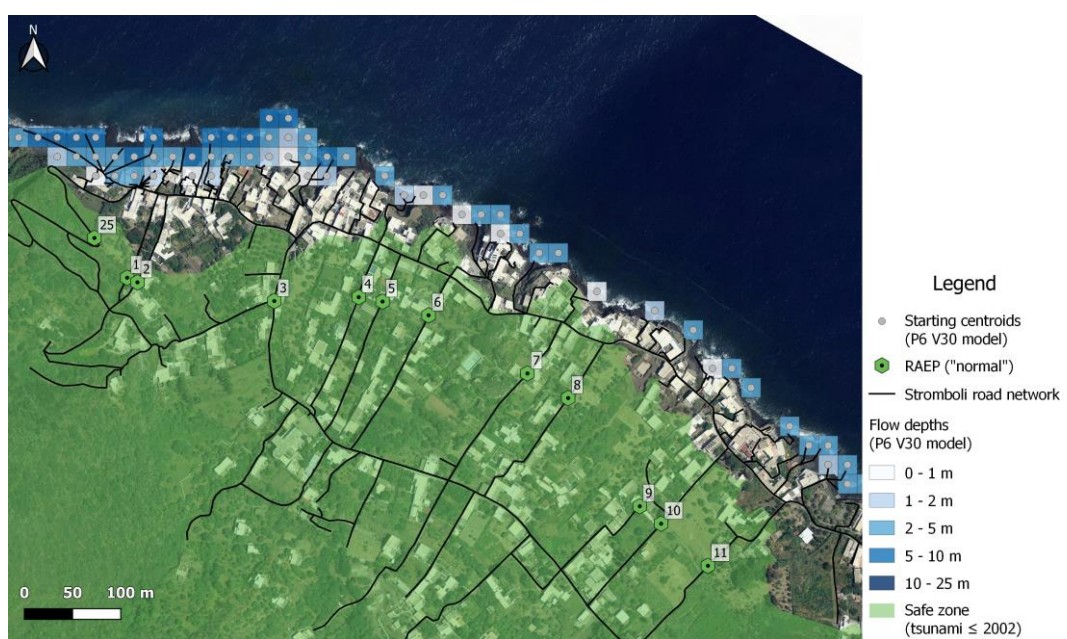

**Figure 2: Establishment of baseline data: starting centroids for the numerical simulation of a 30 million m³ tsunamigenic landslide that detached from position number 6 (294 m below sea level) with a density of volcanic materials of 2.5 kg/m³, Stromboli road network and Refuge Area Entry Point, to run evacuation models. Map was produced by the authors using Quantum GIS version 3.16.7 software (2021).**

**Table 1: Pedestrian evacuation speeds depending on land slope and road types (Péroche, 2016; Bonilauri et al., 2021)**

| Class | Slope value (%) | Associated speed (km/h) | Speeds after application of reduction coefficient (km/h) | | |
|---|---|---|---|---|---|
| | | | Two-lane road | Single-lane road | Passageway, path, stairs, unsurfaced track |
| Reduction coefficient | | | 1 | 0.8 | 0.5 |
| 1 | < 3 | 4.85 | 4.85 | 3.88 | 2.43 |
| 2 | [3–6[ | 4.55 | 4.55 | 3.64 | 2.28 |
| 3 | [6–9[ | 4.26 | 4.26 | 3.41 | 2.13 |
| 4 | [9–12[ | 3.97 | 3.97 | 3.18 | 1.99 |
| 5 | [12–15[ | 3.69 | 3.69 | 2.95 | 1.85 |
| 6 | [15–18[ | 3.42 | 3.42 | 2.74 | 1.71 |
| 7 | [18–21[ | 3.15 | 3.15 | 2.52 | 1.58 |
| 8 | [21–24[ | 2.90 | 2.90 | 2.32 | 1.45 |
| 9 | [24–27[ | 2.65 | 2.65 | 2.12 | 1.33 |
| 10 | ≥ 27 | 1.71 | 1.71 | 1.37 | 0.86 |

For each inundated pixel the difference between the time needed for escape and the wave travel time to the pixel was calculated. In particular, we used two source-to-pixel travel times. Firstly, we used $T_0$, i.e., the onset time of the landslide trigger, then $T_g$, the time of first detection of the generated tsunami at one of the two beacons. It is worth highlighting that the use of $T_0$ gives a longer escape time, but using $T_g$ is more realistic given the current beacon-based alert system.

Finally, a spatial population distribution layer (Figure 3) was created using publicly available data, allowing the creation of three categories of population:

- Population in accommodation, i.e., resident in holiday rentals, hotels and bed & breakfast (B&B) locations. We generally used websites associated with any given establishment, and other websites such as booking.com were used if no direct booking website was available. Capacities for all active establishments were thus taken from as individual rental/hotel/B&B websites, and if unavailable from Booking.com, Airbnb, Tripadvisor, Vrbo, Abritel, Gites.fr or Cybevasion.fr.

- Populations using restaurants, bars, and cafes, with capacities assessed from Tripadvisor and Google images.

- Pedestrian traffic in and out of the port and visitors to beaches. These were taken from our own pictures and head counts made during surveys in September 2022, and in June and September 2023.

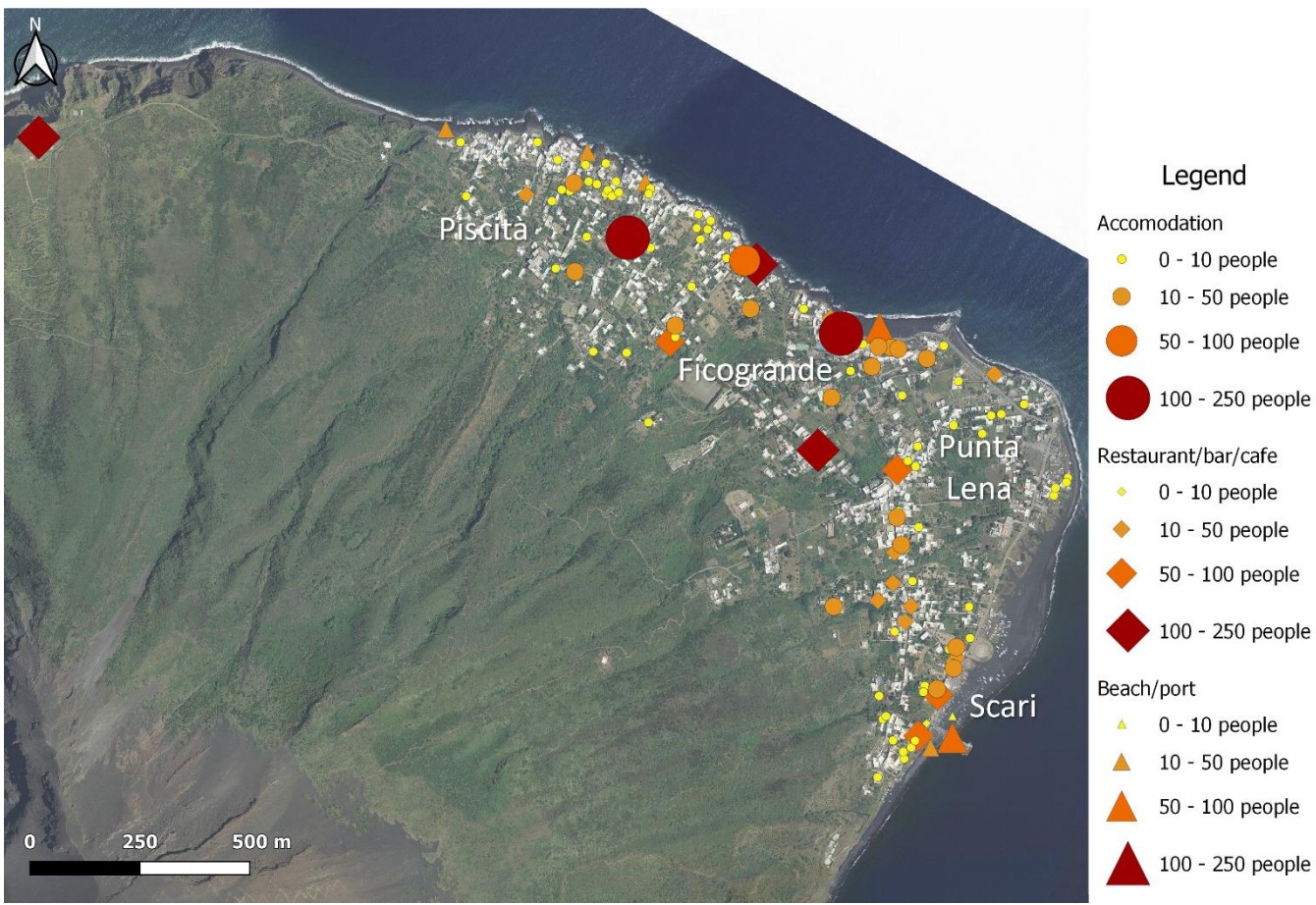


**Figure 3: Spatial distribution of population on Stromboli Island based on Internet public data (accommodation, restaurant/bar/cafe, and beach/port). The authors produced maps by using Quantum GIS version 3.16.7 software (2021).**

For locations where information was not available, we averaged the available capacity of the holiday rentals found and used a density of one person per 12.3 square metres. We then determined occupation scenarios by season and time of day. This
distinguished between winter, when only the permanent population (≤150) is present, and the summer tourist season when as many as 5000 visitors can be on the island. It also distinguished between morning, midday, afternoon, evening, and night between which the distribution of workers and tourists between accommodation, restaurants-bars-cafes, and beaches/port varies.

Combining the vulnerability determined from these surveys with the hazard output for each of the 156 tsunami simulations
provided the number of pixels inundated plus, for each inundated pixel:

- wave arrival time ($T_{arrive}$), where $T_{arrive}$ is the moment when the generated tsunami has reached a pixel and inundation begins to be detected (1 cm threshold). $T_{arrive}$ is calculated from $T_g$;

- inundation depth (i.e., water thickness above ground level);

- evacuation time ($T_{evac}$), where $T_{evac}$ is the moment when evacuees reach a Refuge Area Entry Points (RAEP);

- "evacuability" (= $T_{arrive}$ - $T_{evac}$);

- number of people in need of evacuation.

We statistically convolved these metrics to allow a full and robust, scenario-based risk assessment, which includes generation of probability of inundation and impact score.

## 3 Results

### 3.1 Analysis of individual scenarios: arrival time and coastal impact

The simulation outputs range from tsunamis of just 0.22 m in amplitude, with a run-up of 0.29 m and impacting only some parts of the beaches, to tsunamis with amplitudes of 48.1 m and run-ups of 24.2 m that inundate almost the entire village (Cerminara et al., 2024). For means of demonstration, we take a mid-range example that falls between these two extremes, this being scenario-P6V30CD0.4. This scenario was selected as being slightly larger than 2002, but within the same order of
magnitude, as we already made a detailed analysis of the 2002 events (Bonilauri et al., 2021). Scenario-P6V30CD0.4 involves the simulation of a submarine landslide involving $30 \times 10^6$ m³ of volcanic material with a density of 2.5 kg/m³ from position number 6 (centred at 294 m below sea level). Fornaciai et al. (2019) compared the model output with the extent of inundation and run-up recorded following the 2002 tsunamis and showed that the model favours subaqueous landslide volumes of 15-20 $\times 10^6$ m³ and/or a subaerial landslide of 4-6 $\times 10^6$ m³ on the Sciara del Fuoco. The P6V30CD0.4 scenario used here as the
example thus produces a slightly larger tsunami, with a 12.5 m maximum run up at Spiaggia Lunga where the 10.9 m run up for the 2002 events was measured by Tinti et al. (2006). Such a scenario is likely to occur and could be mitigated for, unlike doomsday end member scenario.

The generated tsunami inundates an area of 544 pixels, i.e., a surface of approximately 217 600 m², with the wave trapped around the entire island (Figure 4a). Refraction of the wave causes some variation in tsunami travel times over short distances,
where arrival times (computed using the trigger time $T_0$ as an initial time) vary from 48 s at Spiaggia Lunga (north of Piscità), to 154 s at Ficograande, and 188 s and 242 s at Punta Lena and the port at Scari, respectively (Figure 4b). These four vulnerable sites are distant from the Sciara del Fuoco by about 2.4, 3.5, 4.1 and 4.9 km, respectively. The arrival times computed using the landslide onset time $T_0$ as an initial time are 20 s longer.

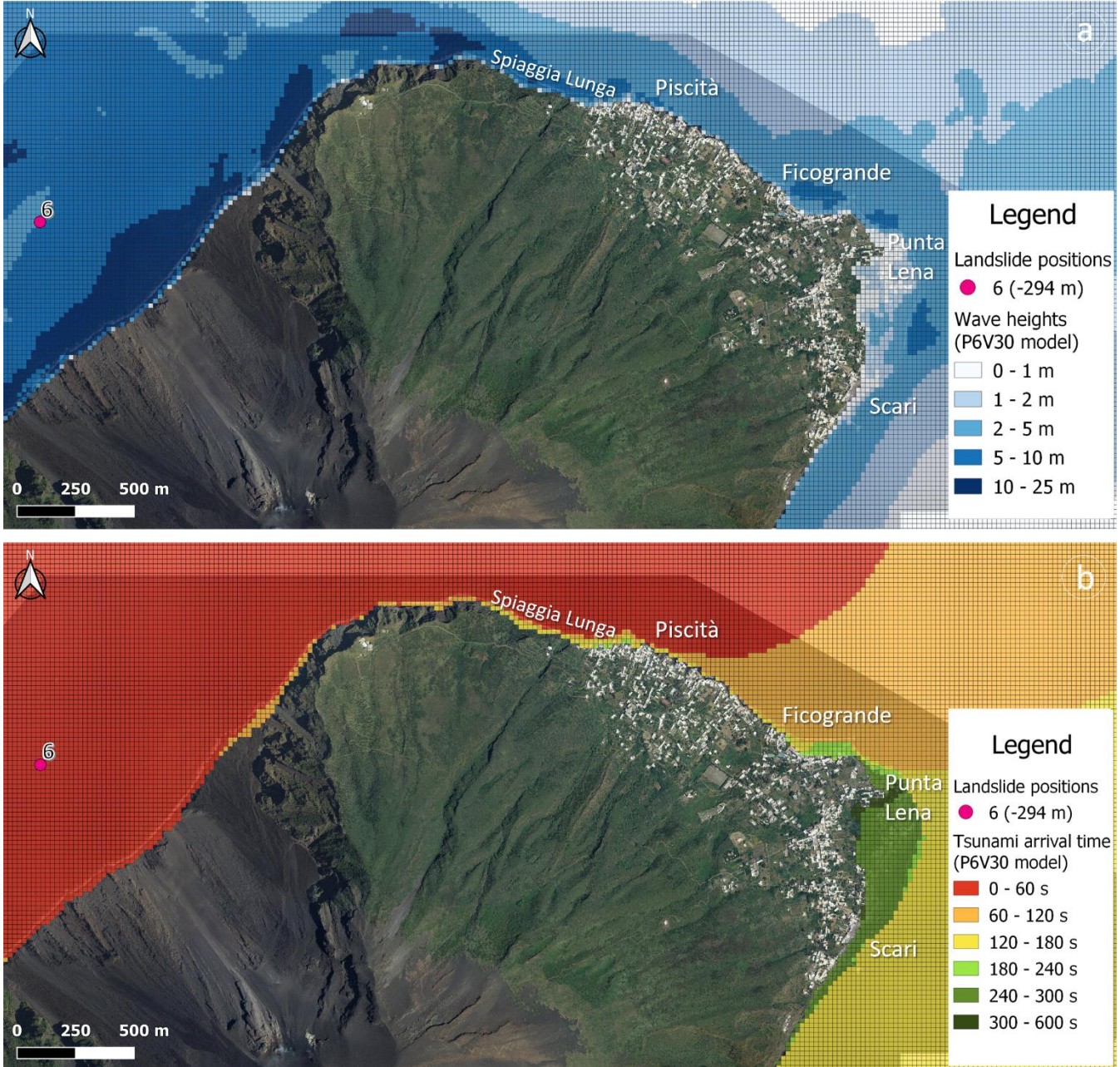

Figure 4: Example of a numerical simulation for a 30 million m3 tsunamigenic landslide that detached from position number 6 (294 m below sea level) with a density of volcanic materials of 2.5 kg/m3. a) Sea level variation around the island of Stromboli due to the propagation of the tsunami wave. b) Time of travel of the tsunami wave around Stromboli. The authors produced maps by using Quantum GIS version 3.16.7 software (2021).

The coastal inundation simulation for scenario-P6V30CD0.4, i.e., the maximum height above ground of water reached by the tsunami in each inundated pixel, is represented in Figure 5a. Inundation depths are highly variable due to the topography of

the village which is underlain by around 11 lava flow fields, erupted from eccentric vents just above the village, between 15 and 2 ka (Calvari et al., 2011; Speranza et al., 2008). To the north of the village while the Spiaggia Lunga is well-exposed, the 20–30 m-high sea cliffs behind it protect the road and building behind it from inundation. The same is true for Piscità, the first district of Stromboli to be reached by the tsunami, which is protected due to its location on the San Bartolo lava flow field. The same presence of the lava flow field along a coastal length of 1 km means that 10 m high cliffs protect the population as far as Ficogrande. Instead, Ficogrande is a bay located at the SE edge of the San Bartolo lava flow field, which focuses the tsunami wave and produces an increase in run-up behind the bay. To the SE of Ficogrande another 0.2 km stretch of coast is protected by the 15m high sea cliffs associated with the San Vincenzo lava flow field. However, beyond this in the southern part of Punta Lena and the northern part of Scari the coastal topography relatively flat coastal for an area extending up to 150-200 m inland, resulting in an increase in the extent of inundation. Refraction of the tsunami wave around the island produces a decrease in the wave height to the south, causing low levels of inundation south of Scari.

The peculiar behaviour of wave propagation around the island impacts the arrival times of tsunamis too. This is particularly true in the low promontory of Punta Lena. Here we observe a narrow, in-land extending band of pixels on the north side of the promontory with an arrival time of 300–600 seconds, while pixels immediately to the south have an arrival time between 120 and 300 seconds (Figure 5b).

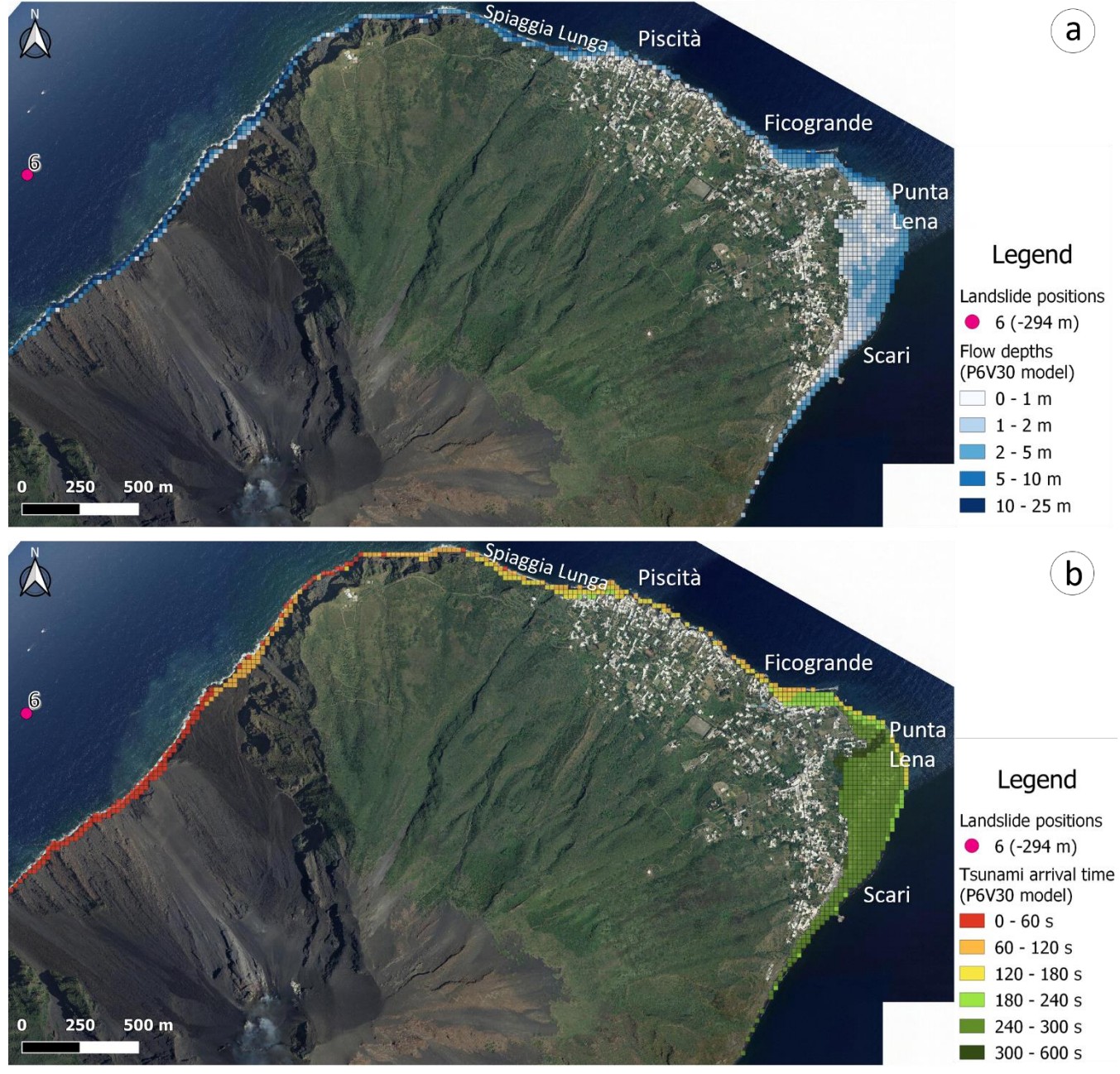

**Figure 5: Example of a numerical simulation for a 30 million m³ tsunamigenic landslide that detached from position 6 (294 m below sea level) with a density of volcanic materials of 2.5 kg/m³. a/ Extent of inundation zone and flow depths. b/ Arrival time of the landslide-induced tsunami on Stromboli coasts. The authors produced maps by using Quantum GIS version 3.16.7 software (2021).**

## 3.2 Impact score

Figure 6 shows the impact score function of landslide position and volume. After no change in the score between positions 0 and 3, the impact score systematically decreases as a function of landslide position between locations 4 and 10 (Figure 6a). The level of impact increases with volume, but with each curve having a similar shape. Subaerial positions (i.e., position numbers 0–3, Figure 1) have higher impact levels, but with little dependence on altitude. In the case of submarine positions (i.e., position numbers 4–10, Figure 1), there impact decreases with depth (below sea level) of the landslide source. For all positions there is a positive, and broadly linear, relation impact and volume (Figure 6b). In the case of the subaerial positions, position 3 which is closest to sea level (Figure 1), shows a slightly higher impact score than the higher elevation subaerial positions 0, 1 and 2. This effect is related to the fact that the sliding volume of volcanic material has not had time to deform before reaching the sea surface, so that its entry is more focused.

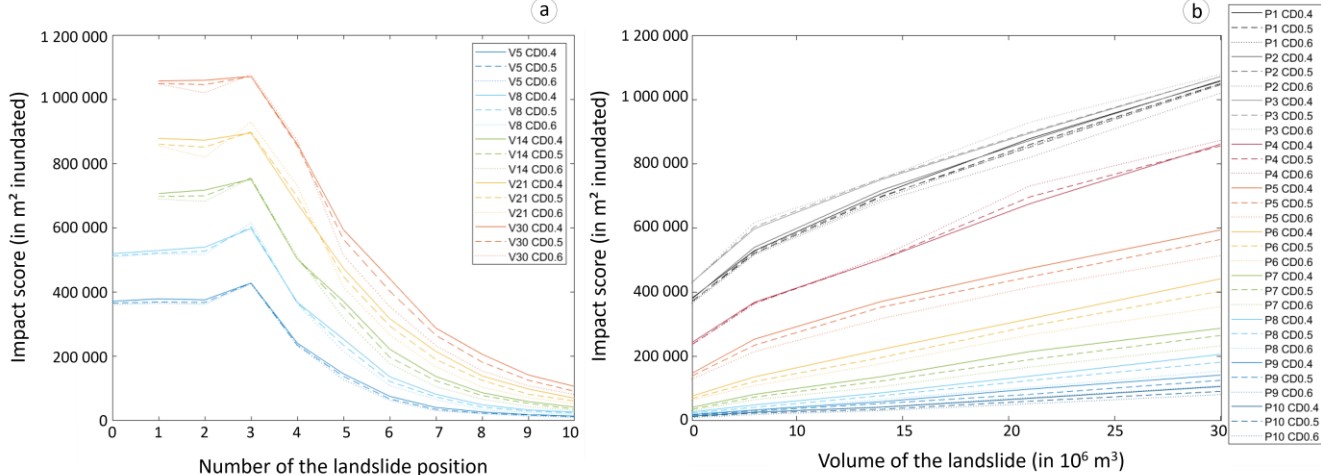

**Figure 6: Evolution of the impact score as a function of a/ position for each volume of volcanic material landslide in the Sciara del Fuoco and as a function of b/ volume for each landslide position (Position 0 to Position 3: subaerial landslide; Position 4 to Position 10: submarine landslide). The authors produced graphs by using MATLAB version 2019b.**

### 3.3 Analysis of the signals at virtual gauges and link with the impact score

The purpose of LASSO is to find the impact score based on the beacon data (without having to analyze the shape of the waves or the volume of the slip from the signal). Our LASSO method is robust in terms of ability to adapt to landslides with volumes of between 5 and 30 million $m^3$, densities of between 1.7 and 2.5 $kg/m^3$ and landslide source positions of between 500 m and - 584 m. That is, within the limits of our simulations. For our simulated beacon signals, we work with landslide models whose smallest volume is $5 \times 10^6$ $m^3$. We are thus not considering the same volume scale as are associated with Pyroclastic Density Current (PDC)-generated tsunamis at Stromboli whose volumes were an order of magnitude smaller than our smallest volume simulation, where the 2019 volumes were estimated by Ripepe and Lacanna (2024) as $2.08 \times 10^5$ $m^3$ (July) and $1.05 \times 10^5$ $m^3$ (August), or the May 2021 event whose volume was estimated at around $8.4 \times 10^5$ $m^3$ by Calvari et al. (2022). Even so, Ripepe

and Lacanna (2024) showed that the shape of the waveform did not change with slip volume, and that it was possible to determine the inundation extent/run-up from the waveform amplitude. Ripepe and Lacanna (2024) also showed that real waveforms matched model-derived (NHWAVE) waveforms, and NHWAVE has been shown through benchmarking to produce results comparable to the model used here (Esposti-Ongaro et al., 2021). We thus have confidence that out synthetic waveforms are valid.

In Figure 7, we link the impact score for each scenario to the absolute value of the wave height. It is worth noting that in the evacuation analysis we used an initial time defined by $T_g$, i.e., the moment when a signal with an amplitude higher than a threshold of 0.3 m is detected. Please note that in the cases in which the signal never reaches this threshold, we assume that the landslide will not generate a tsunami large enough to require an evacuation and will therefore not be considered. Figure 8 shows the wave heights registered at the NE gauge (Figure 7a) and the SW gauge (Figure 7b) in the first 40 s (starting from $T_0$). The first peak is in most cases detected between 20 and 28 s at the NE beacon and between 8 and 16 seconds at the SW beacon. Not surprisingly, we observe that the highest impact scores are always associated with the highest wave heights.

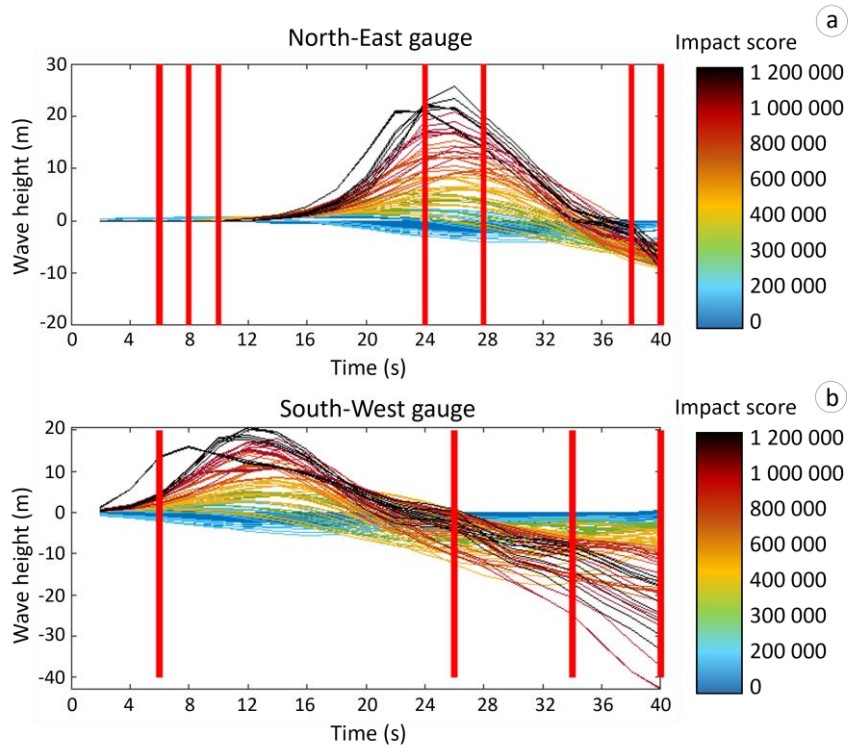

**Figure 7: Gauge signals with a) North-East gauge and b) South-West gauge. The colour code corresponding to the impact score (in m² inundated). The bold red lines correspond to the 11 explanatory variables retained by the LASSO linear regression model.**

To refine this relationship between wave amplitude and impact score, two approaches were adopted (see Methods and Appendix A). The first "rough" approach is a simple linear regression, i.e., we take the maximum amplitude value and relate

it to the impact score (Figure 8a). With this approach some tsunamis with high impact scores are underestimated because the score predicted by the gauge is lower than the score determined by the inundation models (see orange ellipse in Figure 8a). In the second method, the LASSO penalised linear regression algorithm automatically chooses which gauge data values are most likely to explain the impact scores. The method finds 11 "explanatory variables" that best explain the relationship between gauge data and impact scores (Figure 7). As detailed mathematically in Appendix A, these variables are the 11 most critical

times at which a point on the waveform better reproduces the inundation area. The 11 explanatory variables selected were times of 6, 8, 12, 24, 28, 38 and 40 seconds for the NE beacon (Punta Labronzo) and 6, 26, 34 and 40 seconds for the SW beacon (Punta dei Corvi). This is then used to best define and distinguish the specific shape of each given waveform and links it to its inundation capacity. The LASSO regression shows a closer relationship between the gauge data and impact scores than the simple regression and does not underestimate any tsunami with a high impact score (Figure 8b). The LASSO regression

also results in small errors defining a linear relation with a low degree of scatter.

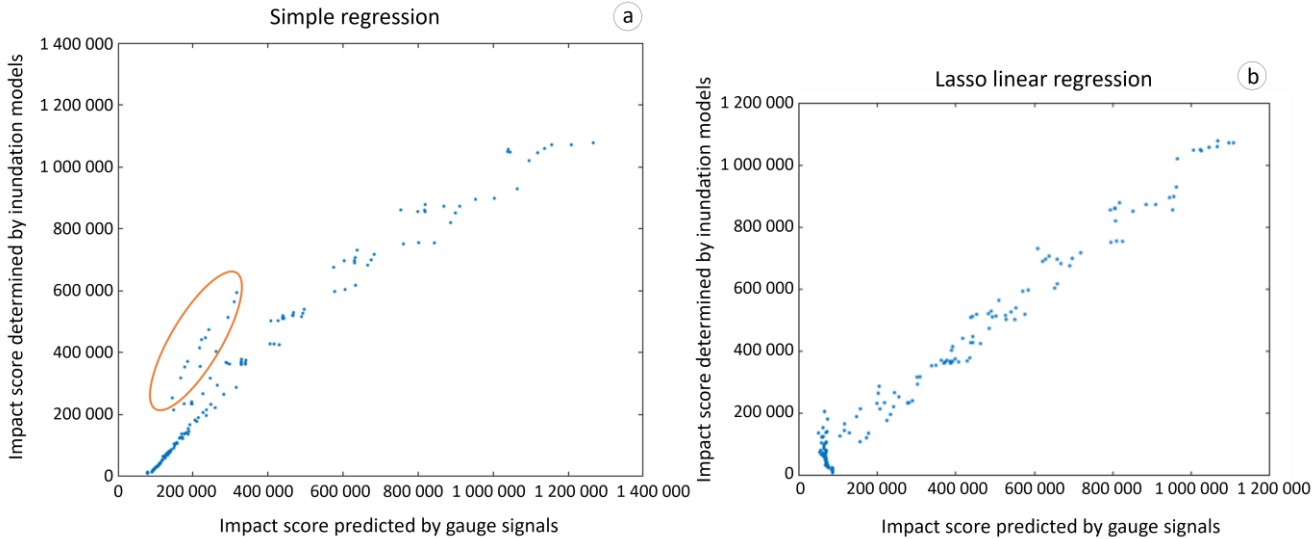

**Figure 8: Inundation expected in m² by tsunami gauge signals according to a) a simple regression approach and b) a Lasso linear regression approach. Blues points represent each simulation of tsunamigenic landslide and orange ellipse represents tsunamis with a high impact score that are underestimated by the approach. The authors produced graphs by using MATLAB version 2019b.**

**3.4 Evacuation capacity**

We here define "warning time needed" and "real warning time". The former is the time needed to move from any given point to a safe point, thus corresponding to the exit from the inundation zone Refuge Area Entry Point (RAEP), while the "real time", is the time available for escape prior to wave arrival. We thus have two types of point, those that for which (1) the RAEP can be reached inside the threshold time and (2) those that cannot be reached inside the threshold time (i.e., outside of threshold time). Threshold

time is, for any given scenario, the wave arrival time minus the time needed to reach the RAEP. If this is negative, the point is outside of threshold time.

Evacuation modelling gives the fastest routes from each inundated pixel to a (RAEP), i.e., an entry point into a safe zone. Results for Piscità, the most proximal community to the source, are given in Fig. 9 by way of example. Results for the Ficogrande, Punta Lena and Scari are given in Appendices 2A, 2B and 2C. As evacuation speeds consider slope and nature of the escape route (path width, surface type, presence of steps, etc.), some routes are not used because, although they appear short they are slow. This explains why RAEP4 and RAEP9 in Piscità (Figure 9), as well as RAEP12 in Ficogrande (Appendix B1), RAEP15 in Punta Lena (Appendix B2), and RAEP22 in Scari (Appendix B3), are not chosen as viable Refuge Area Entry Points.

For each pixel in the inundated zones we used the time for the wave to arrive ($T_{arrive}$) minus the time needed to travel between the pixel centroid and the closest, in terms of time, RAEP (evacuation time, $T_{evac}$), to set the threshold time A positive difference for $T_{arrive} - T_{evac}$ thus means that the pixel can reach a RAEP inside the threshold time for the given scenario, these being the green centroids in Figure 9, and Appendices B1, B2, and B3. Instead, a negative $T_{arrive} - T_{evac}$ means that the pixel reaches a RAEP outside of threshold time, these being the red centroids in Figure 9, and Appendices B1, B2, and B3.

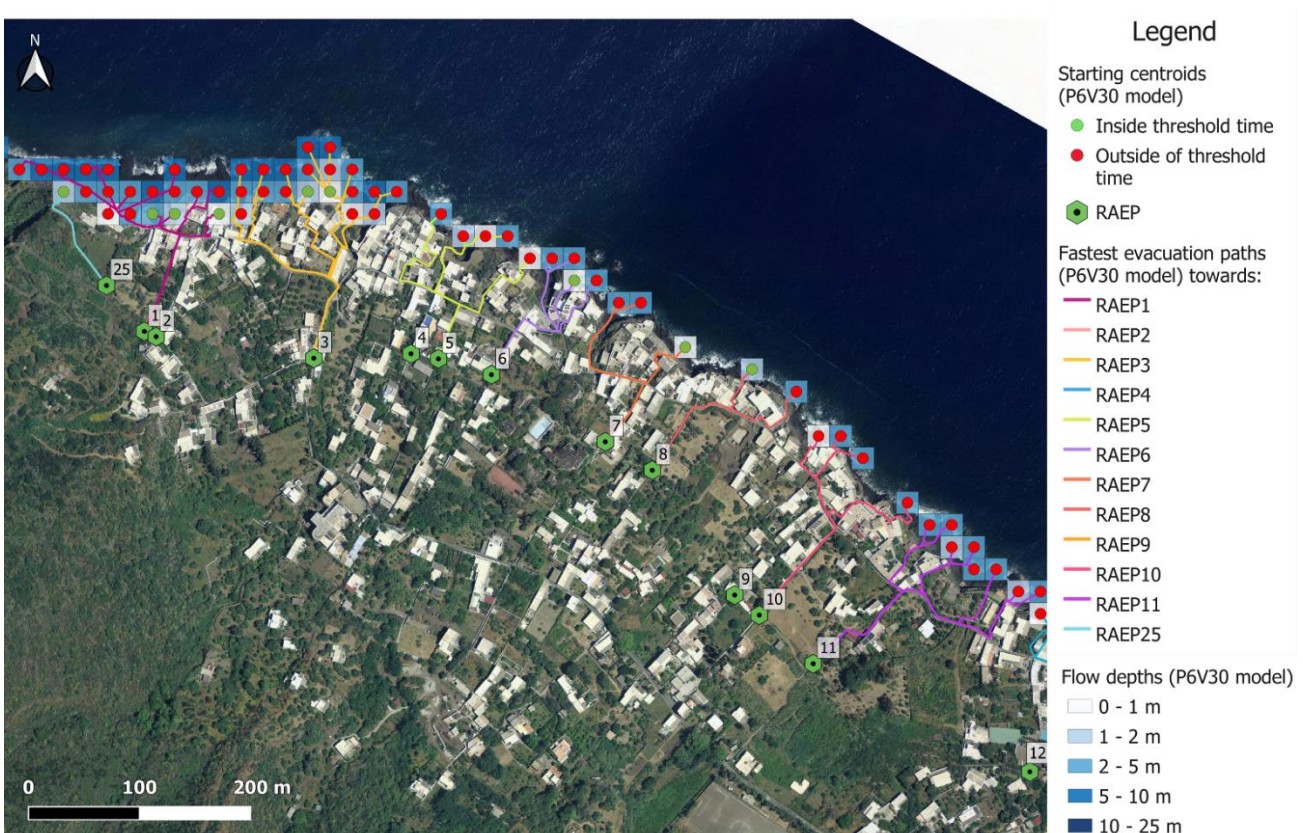

**Figure 9: Example of an evacuation model at Piscità created for a numerical simulation with a tsunamigenic landslide of 30 million m³ having broken away from position 6 (294 m below sea level) with a density of volcanic materials of 2.5 kg/m³. 64 pixels will be inundated (25 600 m²) with their associated flow depth, and the fastest evacuation routes. In green, points are inside the threshold time and in red, points are outside of threshold time. The authors produced maps by using Quantum GIS version 3.16.7 software (2021).**

In our selected simulation example (scenario-P6V30CD0.4), out of 544 inundated pixels (217 600 m²), only 132 (24 %) pixels (52 800 m²) are theoretically inside the threshold time (i.e., the RAEP can be reached before the tsunami arrives). To understand if these pixels can reach a safe RAEP point before the arrival of the tsunami, we compared the real warning times (minimum and maximum for any given zone) available from the tsunami detection gauges and the warning times needed (minimum and maximum) to evacuate the inundated area (Figure 10). In terms of number of inundated pixels, we observe that for Piscità,

Punta Lena and Scari there is time to evacuate some pixels but not the whole area (warning time needed > real warning time). To evacuate the whole area, a maximum of 102 seconds of extra warning time is needed for Piscità, 174 seconds for Punta Lena, and 368 seconds for Scari. For Ficogrande the two curves "warning time needed" and "real warning time" are very close (Figure 10), which means that most pixels can reach a RAEP inside the threshold time (i.e., before the wave arrives). At most an extra 41 seconds are needed to evacuate the whole area.

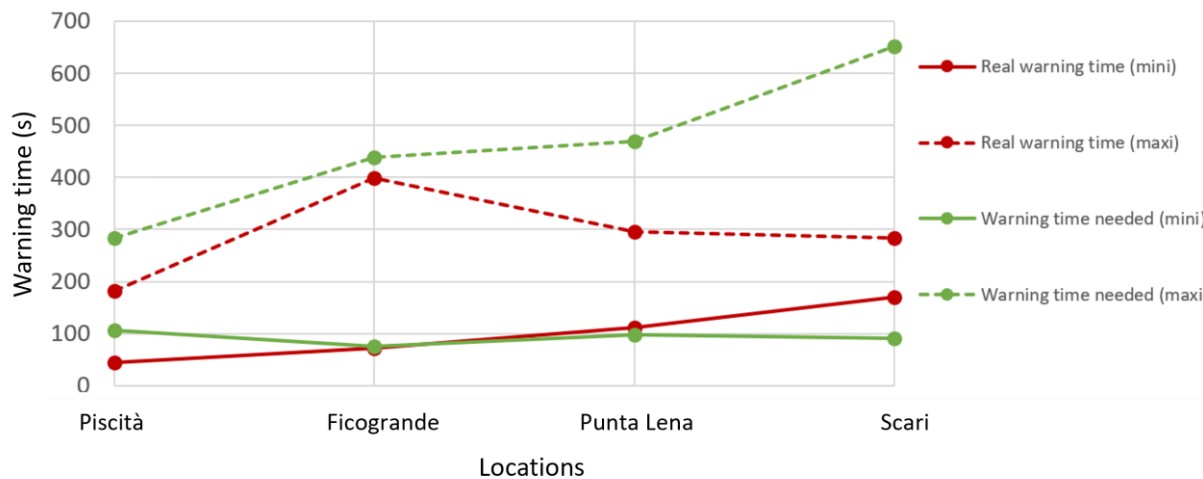


**Figure 10: Minimum and maximum real warning time given by the tsunami gauges and warning time needed for each district to evacuate from the coastal inundated area.**

In Figure 11 we assess the time needed to evacuate 25%, 50%, 75%, and 100% of the population from the inundated zone, and thus the warning time needed to complete partial or full evacuation. In doing this, we need to distinguish between an evacuation

in winter and one in summer. In summer when the coastline if highly populated, differences in the time needed to evacuate 25%, 50%, 75%, and 100% of the population can be observed depending on time of day (Figure 11).

We see in Figure 11 the time of day during which evacuation is most effective varies from district to district. For example, for Ficogrande the best time is day time when tourists gather in locations with easy access to RAEP13. Instead for Scari and Punta Lena all times of day are equally bad due to the long distances to the RAEPs. For Piscità, the situation is slightly better during

the night when people leave the highly exposed beach and gather in the village closer to RAEPs. The situation is exacerbated in Piscità to due to its proximal location and thus the very short wave arrival time.

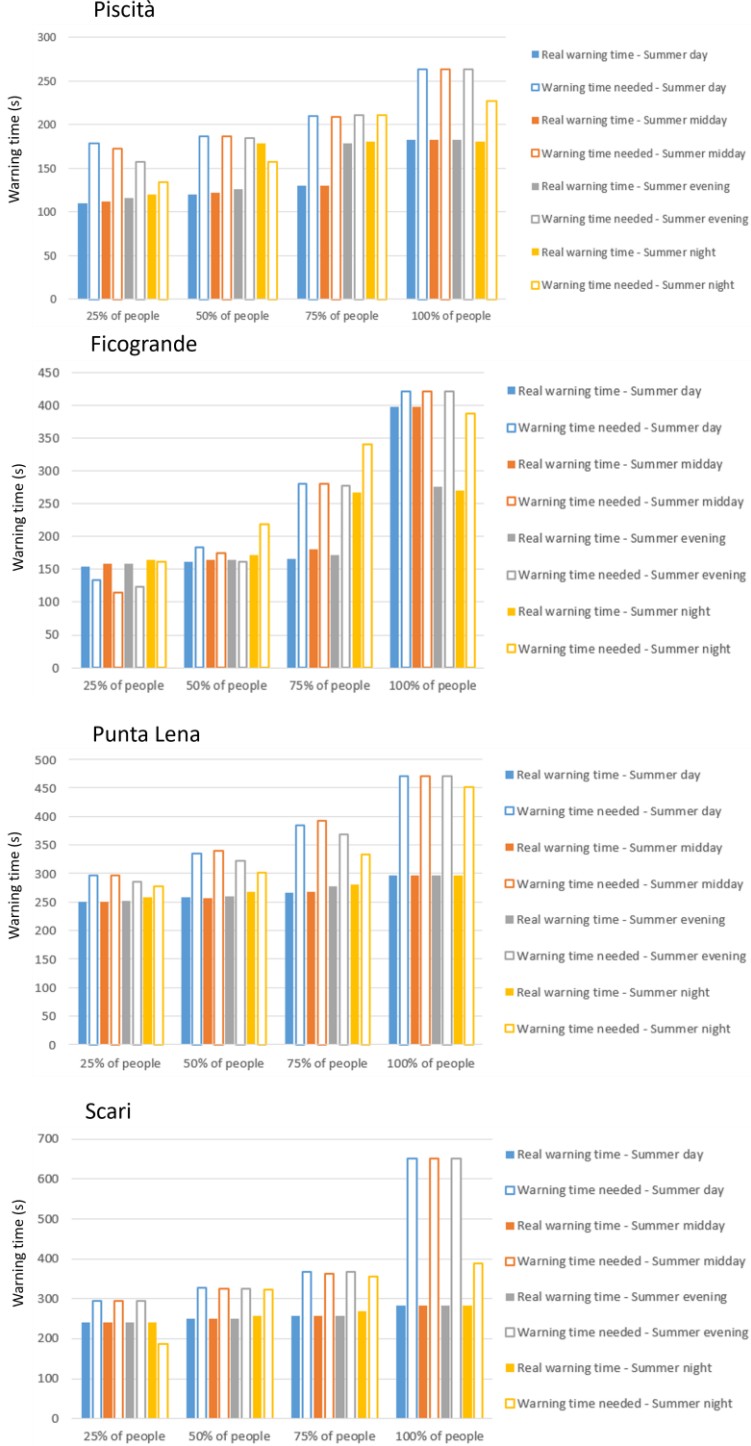

**Figure 11: Comparison between real warning time given by the tsunami gauges and warning time needed to evacuate 25%, 50%, 75% and 100% of Stromboli population from the four main inundated districts. The difference between the filled and empty bars is the number of cases for which we lack the time to evacuate a point for each evacuation scenario.**


To summarize the evacuation capacity of the zone inundated by scenario-P6V30CD0.4, we created histograms giving the distribution between the number of agents that can reach a RAEP inside and outside of the threshold time depending on season and time of day (Figure 12). For this scenario, it is extremely difficult to reach a safe point on time. Out of 544 inundated pixels (217 600 m²), in the best case (a tsunami occurring during a summer night), only 30% of population can reach a RAEP inside the threshold time (Figure 12). We also prepared individual histograms for each of the four main neighbourhoods of Stromboli village (see Appendices C1, C2, C3, and C4). Depending on the district, we can see that during certain periods of the year, and of the day, some are areas not occupied (i.e., their histograms have empty columns). This is particularly the case during the winter period, where a "winter desert" can be found in the districts of Piscità and Ficogrande, which are the locations of summer rentals, B&Bs and hotels. Unfortunately, permanent residents are concentrated in the districts of Punta Lena and Scari, with Punta Lena being a zone with mainly (red) points that reach a RAEP outside of threshold time (Appendix B2). This explains why a winter night tsunami, when the Punta Lena population are at home and close to the shore, the evacuation capacity is 0 % (Appendix C3).

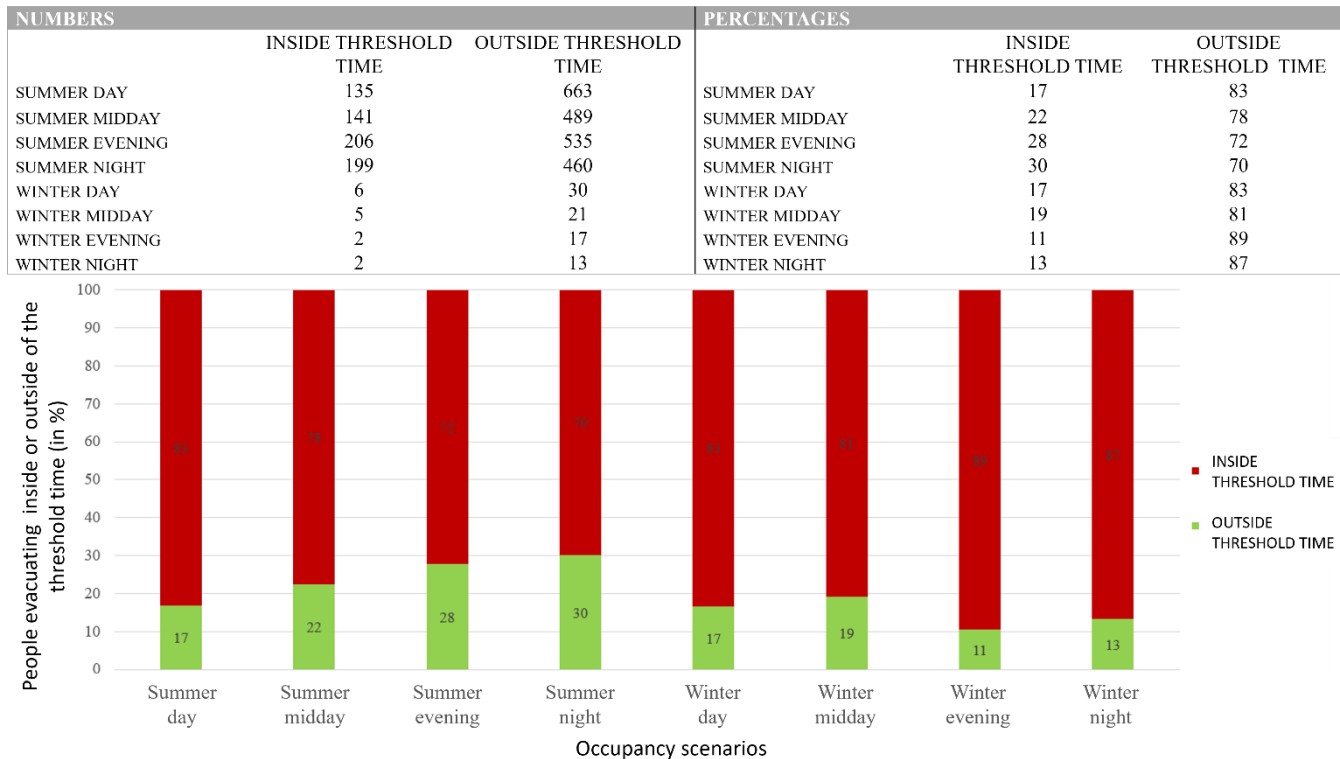

| NUMBERS | | | PERCENTAGES | | |
|---|---|---|---|---|---|
| | INSIDE THRESHOLD TIME | OUTSIDE THRESHOLD TIME | | INSIDE THRESHOLD TIME | OUTSIDE THRESHOLD TIME |
| SUMMER DAY | 135 | 663 | SUMMER DAY | 17 | 83 |
| SUMMER MIDDAY | 141 | 489 | SUMMER MIDDAY | 22 | 78 |
| SUMMER EVENING | 206 | 535 | SUMMER EVENING | 28 | 72 |
| SUMMER NIGHT | 199 | 460 | SUMMER NIGHT | 30 | 70 |
| WINTER DAY | 6 | 30 | WINTER DAY | 17 | 83 |
| WINTER MIDDAY | 5 | 21 | WINTER MIDDAY | 19 | 81 |
| WINTER EVENING | 2 | 17 | WINTER EVENING | 11 | 89 |
| WINTER NIGHT | 2 | 13 | WINTER NIGHT | 13 | 87 |

**Figure 12: Table and histogram of the global number of people who can reach a RAEP inside or outside the threshold time (i.e., before the arrival of the tsunami) in the case of a 30 million m³ tsunamigenic landslide at position 6 (294 m below sea level) with a density of volcanic materials of 2.5 kg/m³.**

## 4 Discussion

### 4.1 Landslide source parameters

The choice of the source input parameters was driven by the need of the Italian Civil Protection to analyse tsunami impacts in
the range of the 2002 event (Tinti et al. 2006b). Larger events are possible (with the upper bound of the $1.81\,km^3$ mass movement for the 5 ka event that formed the Sciara del Fuoco; Kokelaar & Romagnoli, 1995) and they are documented in the geological history of the island (Rosi et al., 2019; Pistolesi et al., 2020).

For what concerns the landslide initial positions, the highest impact is always associated with landslides triggered at the lowest subaerial positions (position 3, which is just above sea-level at 58 m a.s.l). This is due to the fact that granular flows starting
at higher elevations deform and dilute during the downward motion, reducing the front thickness at the impact with water, and thus producing smaller waves. Finally, analysis of the whole database allows us to state that the bulk density of the landslide has a second-order effect on its tsunamigenic capability. Our analysis, carried out on simulation-6 with a density contrast of 0.4, is substantially unchanged for lower densities.

In Appendix D we show the maximum thickness of a $5 \times 10^6\,m^3$ subaerial landslide (measured throughout the entire simulation)
for three different initial positions along the Sciara del Fuoco (positions 0, 2, 3), thus demonstrating that the maximum landslide thickness at the impact with water is higher for the lowest elevation landslide. Conversely, the highest initial position of landslide results in significant lateral spreading and a reduced thickness of the landslide.

We notice in our simulations that, even if the first peak is higher for low-elevation landslides at the proximal gauges, the later peaks are generally of lower amplitude (Appendix E). This might support the idea that the tsunami energy is more
focused on the first peak when the subaerial landslide is less dispersed in terms of lateral extent, but the total energy might still be correlated with the initial landslide potential energy. Verifying such a correlation for granular landslides is thus another of the objectives of our future studies.

In this study, landslide position and volume are considered the key parameters in determining the hazard score (Cerminara et al., 2024). The dependency on other parameters such as landslide initial aspect ratio and horizontal position across the Sciara
del Fuoco, are of second order. Numerical simulations used in this work assume an initial position of the landslide along the central axis of the Sciara del Fuoco (consistently with the 2002 event). This assumption provides an average travel time for the tsunami to Stromboli village (Figure 1). Off-axis landslides will give longer or shorter travel times (by ±30 seconds) for the direction of Stromboli village depending on whether the landslide is in the western (furthest from the village) or eastern (closest to the village) sector of the Sciara del Fuoco, thus affecting the results of the evacuation model. However, this measure
can be very sensitive to the wave height, especially in areas characterized by small slopes or wall bounded regions (e.g., topographic pools). Future numerical investigations will sample initial landslide positions considering potential off-axis sources. Finally, it is worth remarking that different assumptions on the the granular flow rheology and water-landslide friction

might significantly affect the results. Such effects will be analyzed in a dedicated, future work.

## 4.2 Human exposure

For the Aeolian Islands, demographic data are available from the official public census of 2021 (yearly survey by ISTAT, www.istat.it), and of 2011 (www.citypopulation.de). However, data are generally grouped by municipality for ISTAT, and by island and sometimes by district for Citypopulation.de. There are no data for the number of people living in each building, or their demographics. Thus, we had to rely on publicly available data, such as hotel booking forms on the internet, as well as ground-based census completed by us for permanently occupied houses in January 2020 (Bonilauri et al., 2021). Some areas

have therefore probably been over-estimated in terms of population capacity, under the assumption of full capacity for all holiday facilities during the summer season. Ideally, precise population distribution and demographic data would allow us to adjust our evacuation model, but we consider the data used useful in providing a worst-case (full capacity) scenario.

For our evacuation time calculations, we considered a "standard" person with an associated forced walking speed (see Table 1; Péroche, 2016; Bonilauri et al., 2021). These were calibrated and checked as valid during escape simulations carried out by

us in January 2020. Obtaining data on the age and physical abilities of the Stromboli inhabitants would allow construction of evacuation plans tailored to the individuals' capacities (cf. Bonilauri et al., 2021). Our escape times also involve no reaction time: the reaction is immediate, spontaneous, and correct, meaning that the agent escapes immediately using the fastest route.

Provitolo et al (2015) presented two main behaviours in psychological reactions to imminent danger. The first was "instinctive" behaviour where panic or disbelief prevail, and "learned" behaviours where reactions to danger are taken in a reflective manner,

adapted to the context and no longer instinctive. These learned behaviours can be found in preventive or spontaneous evacuations when the context is already known by the population and has been anticipated, and is likely the case for the resident populations (Bonilauri et al., 2021). However, visitors are likely to fall into "instinctive" category. This poses a problem for risk management and mitigation. That is why risk awareness campaigns are organised in volcanic contexts by Italian Civil Protection for several years to reduce reaction times in the face of real danger among the visitor population.

Reaction time is hard to consider because it is specific to each person depending, for example, on knowledge, experience and familiarity with the hazard. Establishing the reaction time of both informed and un-informed agents present in vulnerable areas when the sirens sound is thus our next focus and will allow us to factor this uncertainty into our evacuation models. The same tests will allow us to assess, and refine, the escape times.

In winter, tourism on the island of Stromboli is greatly reduced with visitors numbering less than 5 per day, rising to 20 for

two nights when a visiting football team was on the island (based on our January 2020 survey). Thus, generally, only permanent residents are present in the winter months, which numbered 98 during our January 2020 count. Because most of the population live along the upper roads in the village, under winter conditions only around 20 residents are in vulnerable zones (mostly

Punta Lena). However, these zones are at high risk with only an 11–13 % escape percentage on a winter evening or night (Figure 12), and thus require special attention in terms of evacuation needs.

The real warning time available to us (given by beacons) to evacuate as many people as possible before the tsunami reaches the coast is extremely short due to the short (<5 km) distance between source and point of impact. This means, that even for the best case scenario, only 30% of the vulnerable population can reach a safe point inside the threshold time (Figure 12). As the warning time cannot be reduced, only accelerating the evacuation can guarantee the safety of the people living on or visiting the island. Moreover, the greater the volume of the landslide is, the higher and faster the tsunami generated will be, reducing

the warning time. Thus, this highlights the need to link tsunami waveforms to impact scores, and tailor evacuation times accordingly.

To speed up horizontal evacuation the number of, and ease of access to RAEPs, is necessary, as is distribution of evacuation plans and installation of signage. However, in cases where horizontal evacuation is clearly identified as impossible, vertical evacuation capable of taking the required capacity is the only solution (Bonilauri et al., 2021; Turchi et al., 2022). Since the

response needs to be evacuation, either vertical or horizontal (Leone et al., 2013; Leone et al., 2018; Péroche, 2016; Solís and Gazmuri, 2017), the evacuation plans need to be developed well in advance. Our approach can be used to identify zones suitable for horizontal and vertical evacuation, and be used to set capacity.

### 4.3 Anticipated inundation

The use of the LASSO penalised method (see Method; Figure 8; Appendix A) to link a beacon signal to a predicted inundation

extent is a major advance in our tsunami risk assessment and management. Testing this method on real tsunami detection beacon signals instead of simulated signals would make it possible to determine if the method is valid and reliable in real conditions, and eventually integrating it directly into the beacon algorithms would make it possible to estimate the impact zone before the tsunami reaches the inhabited coast. Currently LASSO needs 40 seconds to recognize and classify the waveform. This time is the result of a payoff between precision in inundation area and time needed to classify the waveform to an

acceptable degree of accuracy. We could reduce this time, but that would make model output decreasingly accurate.  This means that for proximal locations, maps will be delivered after wave arrival, but they will have been alerted by the siren.

Having an estimation on how much coastline will be inundated, and therefore needs to be evacuated, could also aid in disaster planning and management, as well as guiding search and rescue follow-up. The refuge area entry points could be automatically

adapted to each tsunami case by lowering them in elevation for small tsunamis and raising them in elevation for larger tsunamis. In our current study, we considered two types of RAEPs: 15 m a.s.l for smaller tsunamis and 35-40 m a.s.l for larger ones. This would modulate the population in need of evacuation, and regulate the load on escape routes, which could be blocked for agents at risk lower in the zone by agents in the upper zone undergoing an un-necessary evacuation. Such an approach allows

output of the hazard maps and risk assessments in a real time, tailored to the case in hand. We are currently testing such problems using on-site escape tests using multiple agents faced with differing traffic flow scenario's, starting positions (beach, water, bed), and degree of preparation (footware, reaction time). This will enable us to better calibrate our escape times, and to produce a distribution of potential times depending on traffic and route conditions.

## 5 Conclusion

At Stromboli, except the two tsunamis of December 2002, other landslide-related tsunamis have little or no data available. In such cases, hazard scenarios must be supported by modelling, validated by the well-constrained cases. The same can be argued for vulnerability and modelling of mitigation efforts, such as evacuation, where gaps need to be identified and filled through extrapolation. Our integrated modelling approach, which involves a statistical combination of model-output hazard metrics (wave run up, arrival time and sea floor pressure sensor waveform), with a seasonally and diurnally distributed populations plus a model for evacuation times, allows risk mapping to assess whether a point is able of horizontal evacuation or not. The simulated scenarios allow us to consider a wide range of source conditions, enabling for a statistical approach to hazard assessment and providing scenario-tailored evacuation maps. Our approach involves 156 hazard (tsunami) scenarios and five diurnal (morning, midday, afternoon, evening, and night) population distributions divided by season (winter and summer). This results in $156 \times 5 \times 2$ combinations of risk assessment scenarios. Each risk assessment scenario can be linked to the tsunamic type through the waveform recorded by a sea floor pressure gauge, and automatically selected accordingly. Given a warning time of less than a 3–6 minutes on a small volcanic island where the tsunami source is just few kilometres from the vulnerable, shoreline population, such a pre-prepared and automatic event outcome procedure is indispensable.

This method is tested and applied here to the Stromboli, but it can serve as a blueprint for any other volcanic island where volcanic activity and landslides cause local tsunamis with little warning time. Key elements in this are to collect the two most important source terms for hazard modelling (landslide location and volume) as well as spatial and temporal assessments of population distributions. Given uncertainty in the source terms and population distributions our scenario-based approach considers that all simulations have the same probability of future occurrence, as well as a worst case scenario for population distribution. This approach is thus consistent with the precautionary principle of disaster management, while being open to refinement subject to further data collection from, for example, geological studies, expert elicitation, escape tests and/or population surveys, which is our next step.

**Appendix A.** Statistical analysis of tsunamigenic landslides and penalised linear regression LASSO.

Our aim is to define a tsunami impact score based on the coastal inundation, and to understand the link between this score, the associated landslide characteristics, and the gauge signals.

For each tsunamigenic landslide $i$ (i.e., individual $i$) we have:

- The landslide characteristics given by $(V_i, P_i, D_i)$, where $V_i$ is the volume of volcanic material, $P_i$ is the onset position of the landslide along the Sciara del Fuoco, and $D_i$ is the density of the landslide material.

- The area of coastal inundation described by a vector $X_i = (x_{(i,1)}, \ldots, x_{(i,j)}, \ldots, x_{(i,31635)})$ of 31635 values (i.e., the number of pixels of the DEM) of 0 or 1, $x_{i,j} = 1$ if the $j^{th}$ pixel is inundated by the tsunami generated by the landslide $i$.

- The associated gauge signals $BNE_i$ and $BSW_i$, which are two vectors describing the wave heights recorded every 2 seconds since the beginning of the landslide.

For each scenario, the inundation is given by a map of high-dimensional data (dimension corresponds to 31635 values). To reduce the data dimension, we tested three classical statistical methods (PCA, Multidimensional Scaling, and Isomap), and we found that the inundation data is very close to a one-dimensional space characterized by the total number of inundated pixels.

Since we have data for 156 simulations, we used a regression model to reduce the number of scenarios by searching for a relationship between the explanatory variables and the variables to be explained of the $Y = f(X_1, \ldots, X_k) + \varepsilon$, we usually look for the "best" function $f$ within a family of function parameters (e.g., for polynomials of degree less than $d$, then $d$ is the parameter). Due to the high dimensionality (relatively to the number of experiment) we have evaluated two approaches: (1) a rough approach, i.e., retaining only the maximum amplitude on the first major wave detected by each gauge, and (2) a more 505 classical approach which consists in applying a Lasso penalised linear regression to select a smaller number of explanatory variables (Giraud, 2021).

The purpose of a multiple regression is to explain a variable $y$ belonging to $\mathbb{R}$ using p explanatory variables $x_1, ..., x_p$ also in $\mathbb{R}$. In our case, $y$ is the hazard score and $x_1, ..., x_p$ are the signals emitted by the two tsunami detection beacons.

1. The linear regression

It is assumed that $y$ can be determined as a function of $x_1, ..., x_p$ in a linear way, i.e., that there is an affine function $f$ such as:

$$y = f(x_1, ..., x_p) = a_1 x_1 + \ldots + a_p x_p + b + \varepsilon$$

$a_1, ..., a_p$ and b are the unknown coefficients that we will try to estimate from measurements made on individuals. In our case, an individual is a tsunamigenic landslide scenario. The random part ε synthesises the measurement noise and the experimental uncertainties.

For each individual $i$ we measure $Y_i$ the value of the variable $y$ and $X_{1,i}, ..., X_{p,i}$ the values of the variables $x_1$ to $x_p$.

To carry out a linear regression is to look for coefficients $\hat{a}_1, ..., \hat{a}_p$, and $\hat{b}$ which best allow us to estimate $\hat{a}_1, ..., \hat{a}_p$, and $\hat{b}$ from the results of our N individuals.

For this purpose, we define for each individual $i$:

$$\hat{Y}_i (\alpha_1, ..., \alpha_p, \beta) = \alpha_1 X_{1,i} + .. + \alpha_p X_{p,i} + \beta$$

If $\alpha_1 = a_1$, $\alpha_p = a_p$, $\beta = b$ and $\varepsilon = 0$ then $Y_i = \hat{Y}_i (\alpha_1, ..., \alpha_p, \beta)$ and $|\hat{Y}_i - Y_i| = 0$.

We can show that there is equivalence between: "the $\alpha_i$ are all close to $a_i$ and $\beta$ is close to b" and that "all $\hat{Y}_i$ are close to $Y_i$".

This is why the coefficients $\hat{a}_1, ..., \hat{a}_p$, and $\hat{b}$ calculated in a linear regression are those that make the error criterion: $\sum (\hat{Y}_i (\alpha_1, ..., \alpha_p, \beta) - Y_i)^2$ minimum.

Linear regression does not fit the two situations:

- If N (total number of experiments done) ≤ p (total number of explanatory variables), i.e., when there are fewer experiments than the number of explanatory variables.

- Explanatory variables are too correlated.

Our problem lies in both cases so we cannot use classical linear regression.

2. Lasso penalised linear regression

There are ways to get around the limitations of linear regression. One of them is the LASSO penalty. LASSO is described and studied in the book "Introduction to High-Dimensional Statistics" by Christophe Giraud (2021) in which it has been proven that this penalty works and is adequate to solve problems similar to ours.

The LASSO penalty consists in adding to the error criterion of the linear regression, a penalty proportional to the sum of the absolute values of the coefficients. This has the effect of cancelling many coefficients (which means working on fewer
variables and therefore limits the negative effects of the above-mentioned situations).

The criterion to be minimised becomes $\sum(\hat{Y}_i(\alpha_1, \ldots, \alpha_p, \beta) - Y_i)^2 + \lambda(|\hat{b}| + \sum|\hat{a}i|)$.

For each $\lambda$ value we have an associated lasso regression. To choose a $\lambda$ value one can use classical "model choice" methods. For our part, we used the "Leave one out" method thanks to the parameter selection program present in the MATLAB software.

The "Leave one out" method applied to our case gives:

For a $\lambda$:

- For each $i$ (one of our tsunamigenic landslide scenarios),

- virtually remove individual *i* from our experiments (we pretend we have no information for individual *i*);

- apply the LASSO method to calculate the coefficients ($\hat{a}_1$, ..., $\hat{a}_p$, and $\hat{b}$) on all other individuals;

- calculate $\hat{Y}_i*$ the estimated value of $Y_i$ from the previously estimated coefficients then $E_i(\lambda) = (\hat{Y}_i* - Y_i)^2$.

The error associated with lambda is defined by $E(\lambda) = \sum E_i(\lambda)$

 Start again with several values of $\lambda$ and choose the $\lambda$ making $E(\lambda)$ minimum.

**Appendix B.** Examples of evacuation model at Ficogrande (B1), Punta Lena (B2), and Scari (B3) created for a numerical simulation with a tsunamigenic landslide of 30 million m³ released from position 6 (294 m below sea level) with a density of volcanic materials of 2.5 kg/m³. Here we show the pixels that will be inundated and their associated flow depth, and the fastest evacuation routes. In green, points that can reach a RAEP safe point inside the threshold time (i.e., before the tsunami arrives), and in red, points that reach the RAEP outside of threshold time. The authors produced maps by using Quantum GIS version 3.16.7 software (2021).

**(B1) Ficogrande – 79 inundated pixels (31 600 m²)**

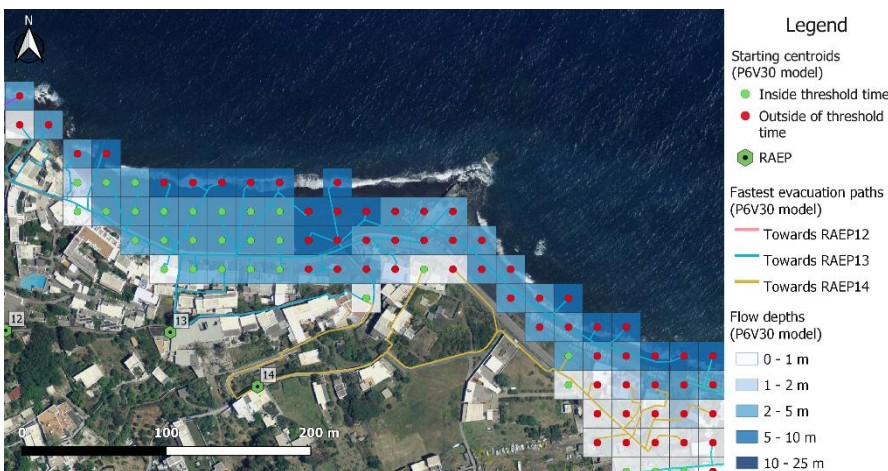

**(B2) Punta Lena – 178 inundated pixels (71 200 m²)**

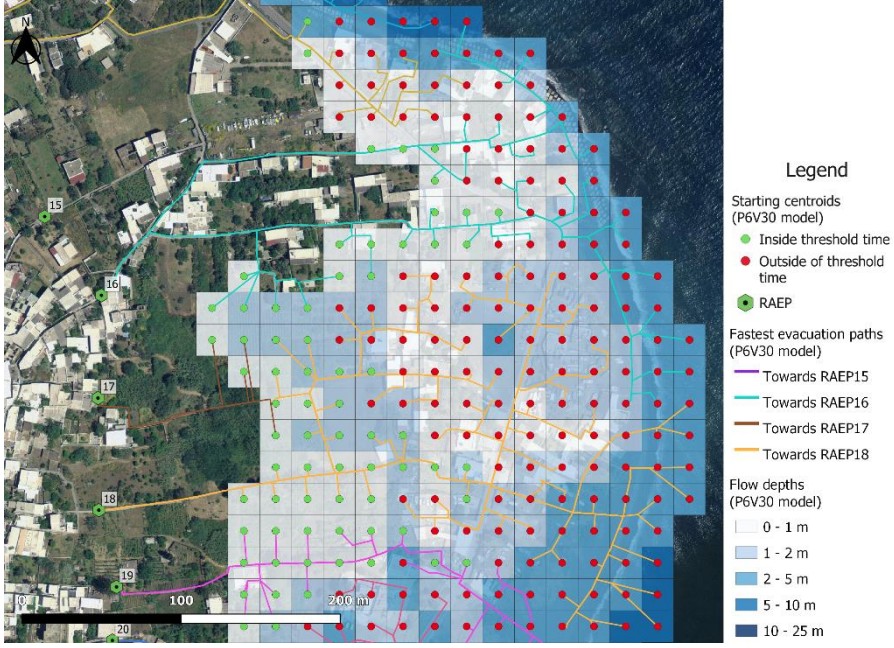

**(B3) Scari – 223 inundated pixels (89 200 m²)**

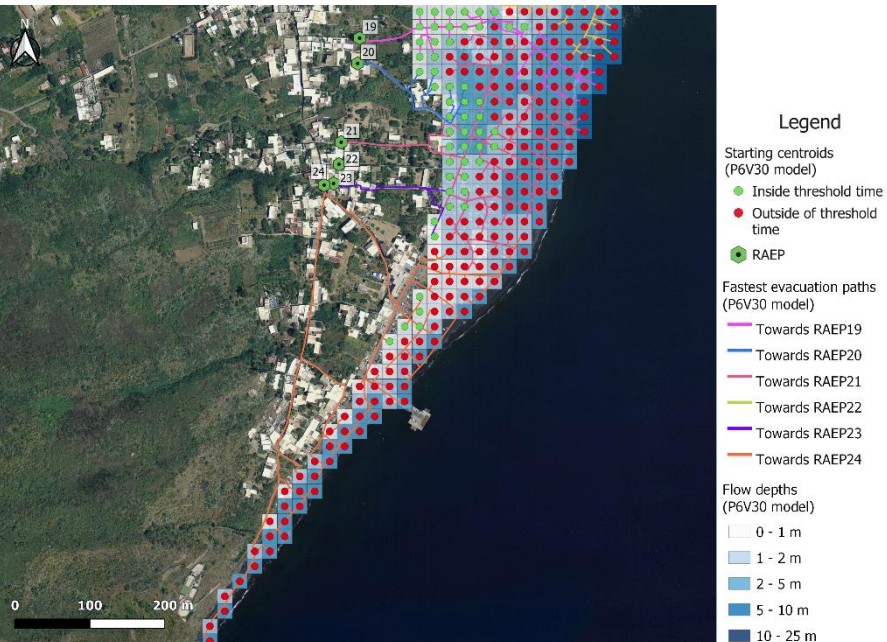


**Appendix C.** Tables and histograms showing the number of people who can reach a RAEP point inside or outside of the threshold time (before the arrival of the tsunami) in the case of a 30 million m³ tsunamigenic landslide (initiated from position 6 at 294 m below sea level, and with a density of 2.5 kg/m³) for four different locations (Piscità C1, Ficogrande C2, Punta Lena C3, and Scari C4).

**(C1) Piscità – 64 inundated pixels (25 600 m²)**

| NUMBERS | INSIDE THRESHOLD TIME | OUTSIDE THRESHOLD TIME | PERCENTAGES | INSIDE THRESHOLD TIME | OUTSIDE THRESHOLD TIME |
|---|---|---|---|---|---|
| SUMMER DAY | 2 | 115 | SUMMER DAY | 2 | 98 |
| SUMMER MIDDAY | 2 | 43 | SUMMER MIDDAY | 4 | 96 |
| SUMMER EVENING | 13 | 58 | SUMMER EVENING | 18 | 82 |
| SUMMER NIGHT | 27 | 41 | SUMMER NIGHT | 40 | 60 |
| WINTER DAY | 0 | 1 | WINTER DAY | 0 | 100 |
| WINTER MIDDAY | 0 | 0 | WINTER MIDDAY | 0 | 0 |
| WINTER EVENING | 0 | 0 | WINTER EVENING | 0 | 0 |
| WINTER NIGHT | 0 | 0 | WINTER NIGHT | 0 | 0 |

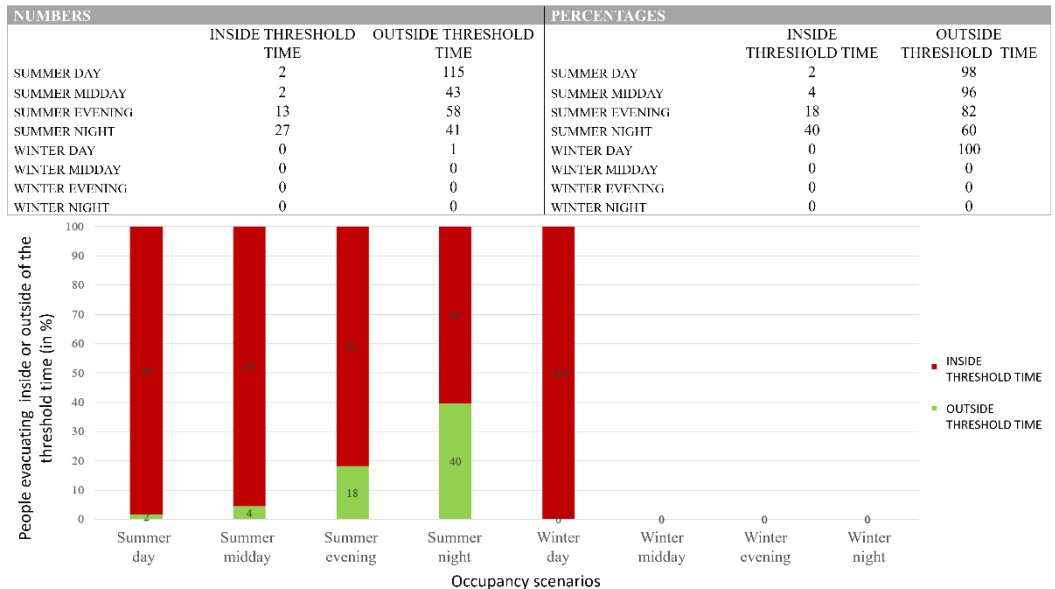

**(C2) Ficogrande – 79 inundated pixels (31 600 m²)**

| NUMBERS | INSIDE THRESHOLD TIME | OUTSIDE THRESHOLD TIME | PERCENTAGES | INSIDE THRESHOLD TIME | OUTSIDE THRESHOLD TIME |
|---|---|---|---|---|---|
| SUMMER DAY | 48 | 78 | SUMMER DAY | 38 | 62 |
| SUMMER MIDDAY | 74 | 88 | SUMMER MIDDAY | 46 | 54 |
| SUMMER EVENING | 119 | 93 | SUMMER EVENING | 56 | 44 |
| SUMMER NIGHT | 78 | 130 | SUMMER NIGHT | 38 | 63 |
| WINTER DAY | 0 | 0 | WINTER DAY | 0 | 0 |
| WINTER MIDDAY | 0 | 0 | WINTER MIDDAY | 0 | 0 |
| WINTER EVENING | 0 | 0 | WINTER EVENING | 0 | 0 |
| WINTER NIGHT | 0 | 0 | WINTER NIGHT | 0 | 0 |

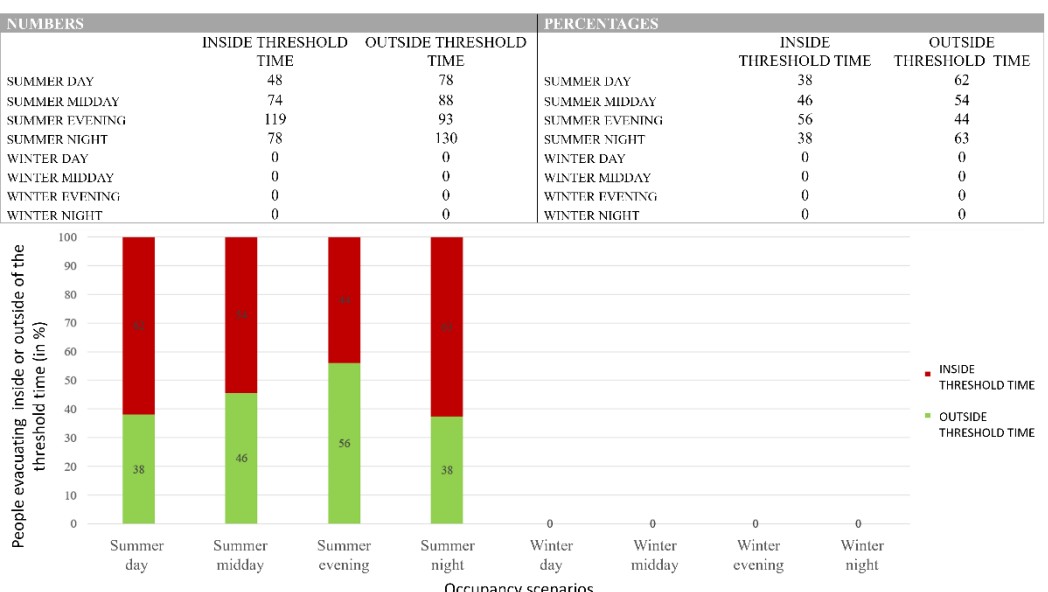

## (C3) Punta Lena – 178 inundated pixels (71 200 m²)

| NUMBERS | | | PERCENTAGES | | |
|---|---|---|---|---|---|
| | INSIDE THRESHOLD TIME | OUTSIDE THRESHOLD TIME | | INSIDE THRESHOLD TIME | OUTSIDE THRESHOLD TIME |
| SUMMER DAY | 28 | 134 | SUMMER DAY | 17 | 83 |
| SUMMER MIDDAY | 25 | 122 | SUMMER MIDDAY | 17 | 83 |
| SUMMER EVENING | 34 | 152 | SUMMER EVENING | 18 | 82 |
| SUMMER NIGHT | 57 | 203 | SUMMER NIGHT | 22 | 78 |
| WINTER DAY | 2 | 10 | WINTER DAY | 17 | 83 |
| WINTER MIDDAY | 1 | 10 | WINTER MIDDAY | 9 | 91 |
| WINTER EVENING | 0 | 6 | WINTER EVENING | 0 | 100 |
| WINTER NIGHT | 0 | 6 | WINTER NIGHT | 0 | 100 |

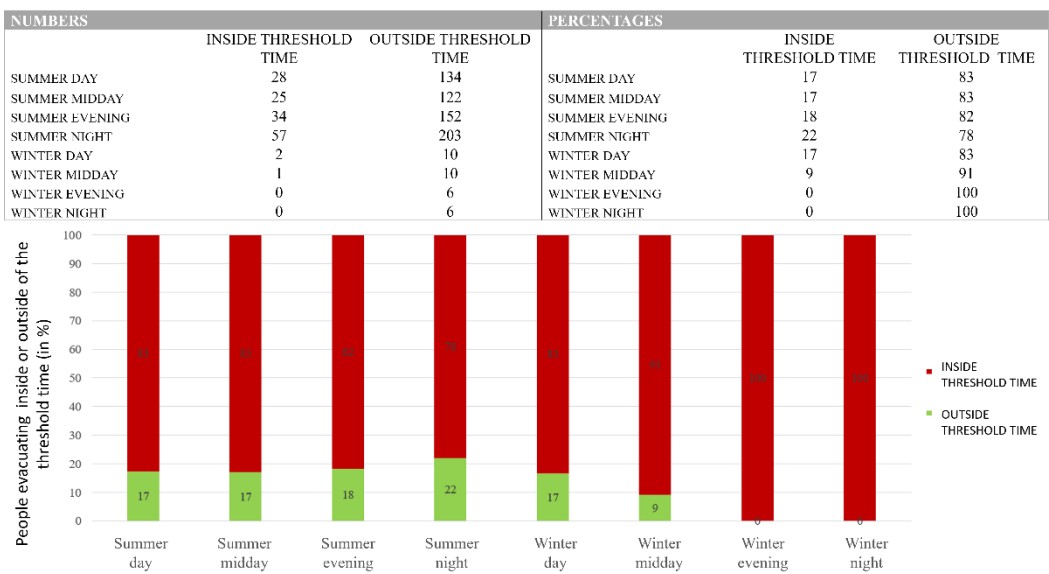


## (C4) Scari – 223 inundated pixels (89 200 m²)

| NUMBERS | | | PERCENTAGES | | |
|---|---|---|---|---|---|
| | INSIDE THRESHOLD TIME | OUTSIDE THRESHOLD TIME | | INSIDE THRESHOLD TIME | OUTSIDE THRESHOLD TIME |
| SUMMER DAY | 57 | 337 | SUMMER DAY | 14 | 86 |
| SUMMER MIDDAY | 40 | 239 | SUMMER MIDDAY | 14 | 86 |
| SUMMER EVENING | 40 | 233 | SUMMER EVENING | 15 | 85 |
| SUMMER NIGHT | 37 | 87 | SUMMER NIGHT | 30 | 70 |
| WINTER DAY | 4 | 19 | WINTER DAY | 17 | 83 |
| WINTER MIDDAY | 4 | 11 | WINTER MIDDAY | 27 | 73 |
| WINTER EVENING | 2 | 11 | WINTER EVENING | 15 | 85 |
| WINTER NIGHT | 2 | 7 | WINTER NIGHT | 22 | 78 |

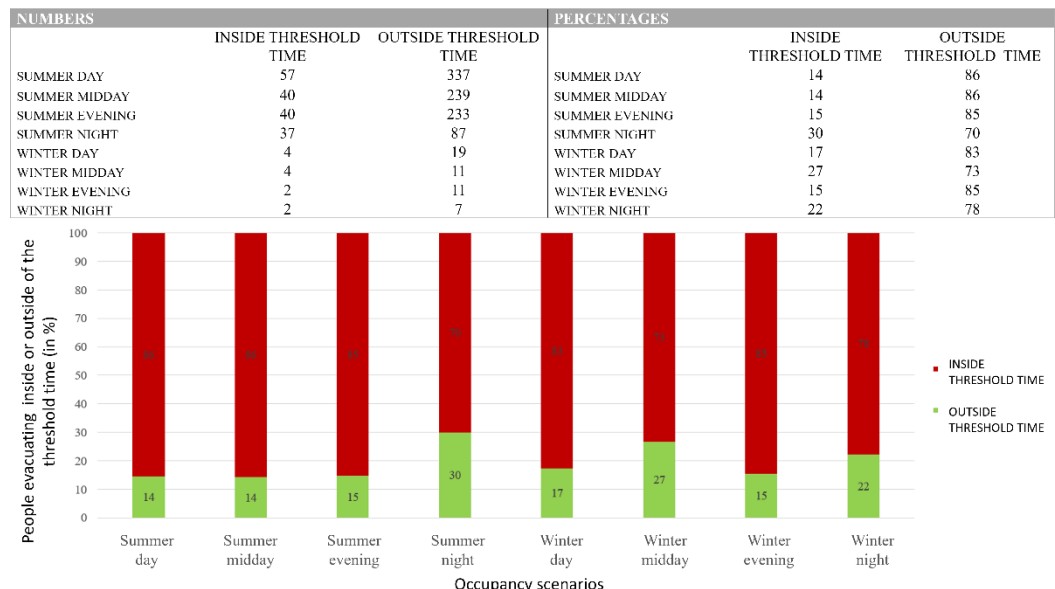

**Appendix D.** Maximum thickness of a $5 \times 10^6$ m$^3$ subaerial landslide from positions 0 (top), 2 (middle), 3 (bottom)
(Cerminara et al., 2024)

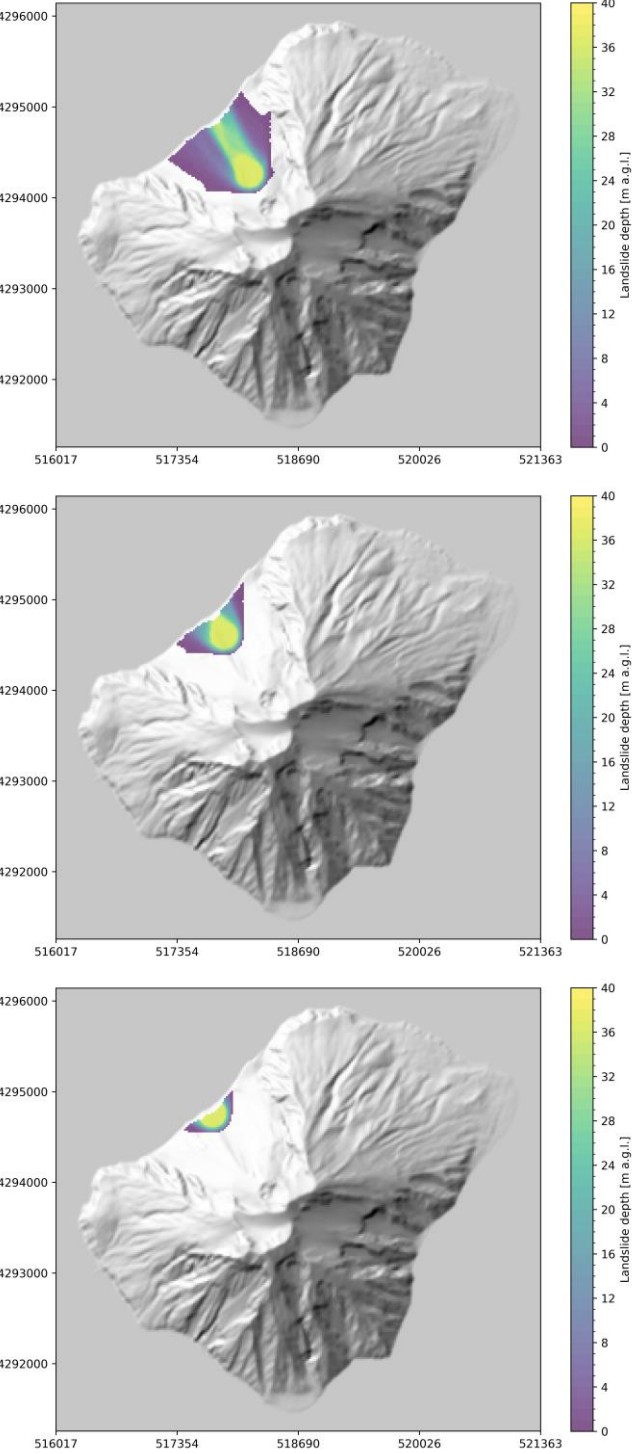

**Appendix E.** Waveforms at the proximal gauge (Punta dei Corvi) for a $5 \times 10^6$ m$^3$ subaerial landslide initiated from positions 0 (purple), 2 (green), and 3 (blue)

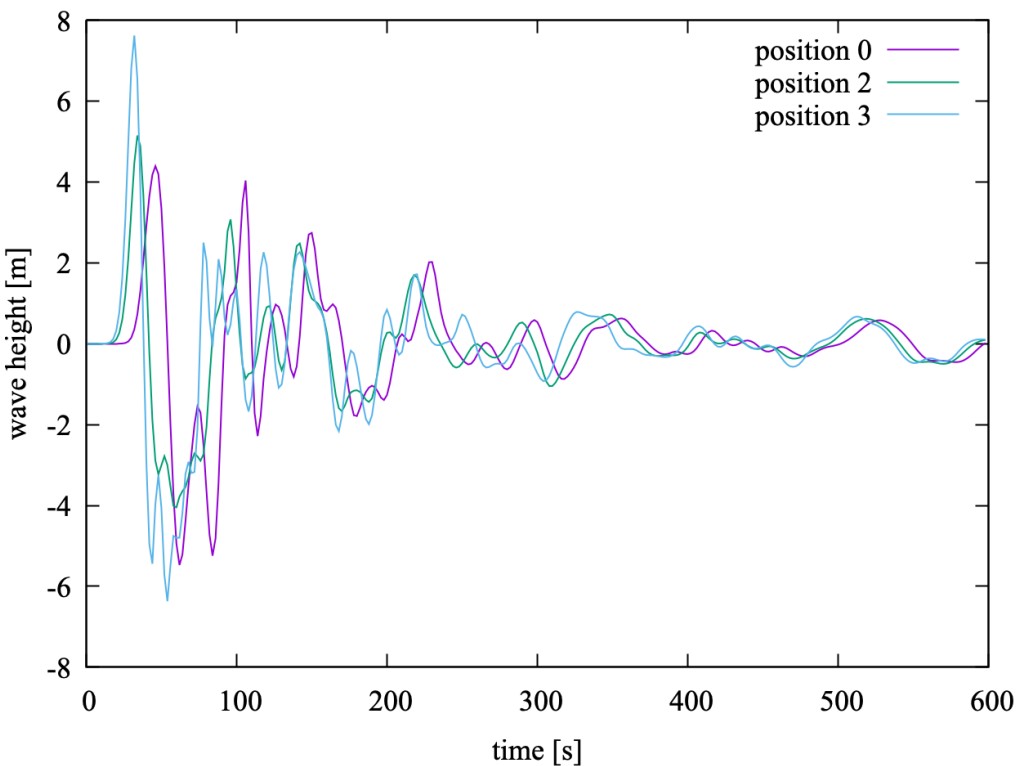

## Availability of data and materials

All maps produced as part of work of Emmie M. Bonilauri are available on request, but the GIS is property of DPC and is thus only available for situations involving officially sanctioned access and collaboration.

## Funding

This work was completed as part of E.B PhD funded by the French government IDEX-ISITE initiative16- IDEX-0001 (CAP 20–25), and under the 2023-2024 Agreement between Laboratoire Magmas et Volcans (France) and Istituto Nazionale di Geofisica e Vulcanologia (Italy).

## Statements and Declarations

This work was supported under the 2012-2021 agreement between Istituto Nazionale di Geofisica e Vulcanologia (INGV) and the Italian Presidenza del Consiglio dei Ministri, Dipartimento della Protezione Civile (DPC), Convenzione B2, WP2 Task 12 (2019-2021), and under the 2022-2025 agreement between Istituto Nazionale di Geofisica e Vulcanologia (INGV) and the Italian Presidenza del Consiglio dei Ministri, Dipartimento della Protezione Civile (DPC), Convenzione attuativa per lo sviluppo delle attività di servizio (2022-2024). This is contribution no. 655 of the ClerVolc program of the International Research Center for Disaster Sciences and Sustainable Development of the University of Clermont Auvergne.

## Authors' contributions

EB was responsible for all data processing and analysis through set up of the GIS and its data layers. MC, BC, TEO completed all tsunami modelling and provided the hazard data layers. CA guided development and application of the statistical methodologies. RP and AH conceptualised designed and guided the work. DM ensured the results were applicable to the field needs of Civil Protection. EB led manuscript figure and table preparation with input from all co-authors.

## Competing interests

The authors declare that they have no competing interests.

## Acknowledgements

E.B acknowledges the support of the French government IDEX-ISITE initiative16- IDEX-0001 (CAP 20–25) and the whole group acknowledge the support from Italian Civil Protection through the 2022-2025 agreement between Istituto Nazionale di Geofisica e Vulcanologia (INGV) and the Italian Presidenza del Consiglio dei Ministri, Dipartimento della Protezione Civile (DPC), Convenzione attuativa per lo sviluppo delle attività di servizio (2022-2024), and the 2023-2024 Agreement between Laboratoire Magmas et Volcans (France) and Istituto Nazionale di Geofisica e Vulcanologia (Italy).

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
