# Peer review of "Inundation and evacuation of shoreline populations during landslide-triggered tsunami: An integrated numerical and statistical hazard assessment"

_EGUsphere, 2024_

## Author Comment (AC1)

Comments of Reviewer 1 - our replies are in red

MS No.: egusphere-2024-221

MS type: Research article

Please find below my evaluation for the manuscript "Inundation and evacuation of shoreline populations during landslide-triggered tsunami: An integrated numerical and statistical hazard assessment" by Emmie M. Bonilauri et al. for consideration in NHESS.

The Authors present a novel study which combines landslide-triggered tsunami modelling at Stromboli volcano (Italy) with evacuation procedures and their potential effectiveness. The combination of the two aspects allows the authors to test escaping routes, timing needed for evacuation, efficacy of evacuation based on the different time of the day and of different seasons. The manuscript is well written and well designed, accompanied by clear figures. A pleasure to read.

I have a very few concerns regarding the structure or the way topics are investigated, which, if addressed, hopefully may result in higher clarity of the text.

- The first point is related to the choice of the case (i.e. wave height) for which results are actually presented. Whereas in methods, five different volumes (5, 8, 14, 21, and $30 \times 10^6$ m$^3$) are indicated to be used with the modelling, the case used just for data presentation is considered a medium case in terms of wave amplitude (which varies 0.22 to 48.1 m as actual total range). By means of demonstration, the mid-range example that falls between these two extremes is scenario P6V30CD0.4, a simulation of a submarine landslide (P6) involving $30 \times 10^6$ m3 of volcanic material. So, a $30 \times 10^6$ m$^3$ is the maximum used volume for the simulations, and the average example in terms of wave amplitude is obtained when this detaches from P6. I thus assume that the maximum waves with a double amplitude (c.ca 40 meters) result from similar volumes but linked to different conditions (height of collapse, maybe P3). Given that this volume is comparable to the 2002 event, is there a correlation with observations? How is this number compared to the Fornaciai et al. (2019) in the case of a submarine slide? Apparently their run-up is smaller (see their figure 4 for run up 20 million cubic m) or figure 3 (for wave amplitude at the beacons).

This scenario was selected as a "probable" scenario for the near-future, i.e., a wave slightly larger than 2002 but within the same order of magnitude, as we have already made a detailed analysis of the 2002 event (Bonilauri et al., 2021). We now state this in the text (Section 3.1, lines 207 – 208).

The article by Fornaciai et al. 2019 compares the model output with the extent of inundation and run-up recorded following the 2002 tsunamis and shows that the models favour subaqueous landslide volumes of about 15-20 $\times 10^6$ m$^3$ or/and a subaerial landslide of about 4-6 $\times 10^6$ m$^3$ along the SdF. The P6V30CD0.4 scenario used as an example in the results section is a scenario producing a slightly larger tsunami (12.5 m max run up at Spiaggia Lunga where the 10.9 m run up was measured by Tinti et al 2006) than those of 2002 but in the same order of magnitude, so that it can be managed on the ground by civil protection. This scenario is therefore likely to occur and will require the population to be evacuated. We have clarified this in the text (Section 3.1, lines 210 – 217).

- The correlation among amplitude at the beacons and potential inundation area is the weakest point to me. First, while a correlation (with the LASSO approach) is clear, I did not get the point of how the different waveforms were treated (with the submarine landslides I would have expected first negative waves). Second, while a relation is visible, it is speculative in a way that all the synthetic data is not tested with at least a single real case, even for what concerns at least the first part of the process (i.e. relation among landslide volume and wave amplitude). Though beyond the scope of the work, this would reduce a variable and, to my knowledge, wave amplitudes (from the beacons) and landslides volumes (from GIS data) are available for the 2019 and 2021 observed PDCs events (see Ripepe and Lacanna 2024 for the beacons, Calvari et al. 2022 for the PDCs volume).

For the first point, we now better explain the purpose of LASSO, which is designed to find the impact score based on the beacon data (without trying to analyse the shape of the waveforms or finding the volume, position or density from the signals) (Section 3.3, lines 267 – 270).

For the second point, we did not have enough real landslide data to produce reliable statistics, so switching to simulations have been carried out by the INGV-Pisa group (Fornaciai et al., 2019; Esposti Ongaro et al., 2021; Cerminara et al., 2024). Position 0 (altitude: 500 m), which has been added to the initial database of the simulations, corresponds to the subaerial landslide position of the December 2002 events. The simulated inundated area with the physical characteristics of 2002 corresponds to what was observed in the field (Fornaciai et al., 2019; Esposti Ongaro et al., 2021).

For the last point, our simulated beacon signals, we work with landslide models whose smallest volume is $5 \times 10^6$ m$^3$. We are not considering the same volume scale as the July and August 2019 PDCs, whose volumes were an order of magnitude smaller than our smallest volume simulation (volumes of $2.08 \times 10^5$ m$^3$ and $1.05 \times 10^5$ m$^3$ respectively as estimated by Ripepe and Lacanna (2024), or the May 2021 event whose volume was estimated at around $8.4 \times 10^5$ m$^3$ by Calvari et al. (2022).

Ripepe and Lacanna stated that "This volume (i.e., that of the 2021 events) is well in harmony with the $0.8 \times 10^5$ m$^3$ volume of material collapsed from the north flank of the NE crater estimated from images taken by helicopter immediately after the failure (Calvari et al. 2022)." … but in fact Calvari et al. (2022) gives $8.4 \times 10^5$ m$^3$.

However, Ripepe and Lacanna 2024 showed that the waveform did not change with the slip volume and that it was possible to determine the inundation extent/run-up from the tsunami amplitude, which we also show here. But indeed in the future, in the case of a major event with a volume greater than 5 million m$^3$, we will be able to test our LASSO penalised linear regression technique on real signals. We now explain all of this in the text (Section 3.3, lines 270 – 279).

Notes on Ripepe and Lacanna (2024):

"We show how waveform and period of the tsunamis do not change with the landslide volume nor seems to be affected by landslide cinematics."

"Besides, the tsunamis simulated by NHWAVE numerical modelling (see Fig. 3finref.28) have a waveform remarkably similar to the tsunamis recorded both on July and August 2019 (Fig.5a)."

"This suggests that a linear relationship (V = 6.8x10$^5$ . A − 3.9x10$^5$) between tsunami height (A) and landslide volume(V) can be considered reasonably acceptable, with implications on our ability to promptly assess the hazard along nearby coast."

"Inundation scenarios assuming different sliding volumes could be, in fact, pre-calculated and used to relate the amplitude of the tsunami detected by the gauges to the effects on the nearby coasts in almost real-time"

- Although modelling is not the main core of the paper, a more thorough discussion on the physics of the phenomenon would help in better understanding why position 3 is able to generate the largest amplitudes, even larger than higher positions (0, 1, 2). An explanation on the modification is given in the discussion (flow modification during the flow), but to me this is a major point that should be expanded a bit. I recall again the crater collapse events of 2021, and I wonder if these have occurred with the same volumes close to the shoreline, one may have expected higher waves. Is it a matter of initial state of the material (i.e. solid block vs loose material)?

We thank the reviewer for the remark, that allows us to explain in more detail our results. In our simulations, we observe that the highest wave peaks (that, for subaerial landslides, is always the first one) are always associated with lower elevations of the landslide, and we have explained this behaviour by invoking the different behavior the deformable granular material and thus the front thickness at the entrance of the sea.

The following figures (Figure 1) show the maximum thickness of a 5 Mm$^3$ subaerial landslide (measured throughout the entire simulation) for three different initial positions along the Sciara del Fuoco (positions 0, 2, 3). The figures demonstrate that the maximum thickness of the landslide at the point of water impact is higher for the lower elevation of the landslide. Conversely, the highest initial position of the landslide results in a significant lateral spreading and a reduced thickness of the flow. These effects are attributed to the granular behaviour of the landslide on the SdF slope. We now add this to the discussion (Section 4.1, lines 390 – 393) supported by a new appendix, Appendix D.

[Figure]

Figure 1. Maximum thickness of a 5 Mm$^3$ subaerial landslide from positions 0 (left), 2 (center), 3 (left)

We notice in our simulations that, even if the first peak is higher for low-elevation landslides at the proximal gauges, the later peaks are generally of lower amplitude (Figure 2), as also added to the Discussion (Section 4.1, lines 394 – 395) and supported by a new appendix, Appendix E.

[Figure]

Figure 2. Waveforms at the proximal gauge (Punta dei Corvi) of a 5 Mm³ subaerial landslide from positions 0 (purple), 2 (green), 3 (blue)

This might support the idea that the tsunami energy is more focused on the first peak when the subaerial landslide is less spread, but the total energy might still be correlated with the initial landslide potential energy. Verifying such a correlation for granular landslides is one of the objectives of our future studies. Again, added to the Discussion (Section 4.1, lines 395 – 398).

- A minor point is related to the number of population in accommodation obtained from maximum capacity given in the web (Booking.com, Airbnb, Tripadvisor etc.). Maybe it is a stupid question, but I am wondering if the availability given in booking.com is only a part of the total availability for each single structure (they common allow online reservation only for a part of the rooms). Can this results in an underestimation of total holiday touristic capability? I also suspect that the Scari pier population distribution should be higher (in the range of 100-250) given that during summer days the pier is full of persons waiting for the boats during some periods of the day.

We generally went to the direct site associated with any given establishment, and booking.com if no direct website was available. We agree, though, that we probably have a slight underestimate. As now stated (Section 2.3, lines 174 – 178).

Improving our knowledge of the population distribution by time of day, day of the week and time of the year is something we are actively working on. This includes examining boat capacities and logging arrivals and departures on site, as well as beach usage. We note that many visitors may just stay for a few hours in the afternoon.

- Another curiosity is related to the waves arrival times and the discussion on submarine and subaerial lava flows. While the effect of subaerial lava flows is discussed, I am wondering if the effect of submarine lava flows can play a role and if this has been tested in some way.

In this paper, we only considered landslide. Lava flows and PDC would no doubt have a different interaction dynamic which is unfortunately beyond the scope of our model and analysis.

- Another point is related to the 11 explanatory variables of the LASSO approach that are only mentioned in the main text. At least they should be briefly summarized (not only in the supplementary material, where however I could roughly get an idea of which kind of variables were tested).

We now explain these in the text (Section 3.3, lines 296 – 300), but prefer to leave the pure mathematical explanation in the Appendix. However the explanatory variables selected were times of 6, 8, 12, 24, 28, 38 and 40 seconds for the NE beacon (Punta Labronzo) and 6, 26, 34 and 40 seconds for the SW beacon (Punta dei Corvi). We have added this to the text.

- A final point is related to the crowd effect. Despite this is not an agent-based evacuation approach, I am wondering how the fastest pedestrian evacuation paths from a danger point to a safe point may be affected by a huge number of persons in the small routes at the same time. Can we say that this is the best-case scenario, i.e. single person escape time and that this could largely increase when a lot of people are together in the small routes?

As with the visitor distribution, this is something that we are actively working on. We are currently finalizing tests using agents in the field to assess escape time under different "traffic" conditions, as well as footwear, location (in bed, in the water …) and response time. We are also testing multi-agent models to convolve with our current 2D approach. We have highlighted this at the end of the discussion (Section 4.3, lines 470 – 473).

FIGURES

All figures all well designed. I guess there is no need to specify "Created by INGV-Pisa". I would add some toponyms (i.e. Spiaggia Lunga) used in the text for clarity. The figures have been modified in line with your recommendations.

---

## Author Comment (AC2)

Comments of reviewer 2 - our replies are in red

Reviewer report for 'Inundation and evacuation of shoreline populations during landslide-triggered tsunami: An integrated numerical and statistical hazard assessment' by Emmie M. Bonilauri et al.

**General Comments**

This paper studies a very challenging problem, that of whether and how residents and visitors to Stromboli can be evacuated to safety in time after a landslide on the flanks of the volcano causes a tsunami. On the whole this is a good-quality paper, presenting modelling of landslide-caused tsunamis at the volcano, methods for their detection and quantification, and estimates of which of the areas at risk can potentially be evacuated in time.

The biggest problem I found in reading this paper was with understanding Section 3.4 on Evacuation Capacity, where it was difficult for me to interpret as some of the key metrics were not clearly enough defined, which made it difficult to understand some of the figures and to confirm some of the stated implications. I will put more specific comments below, but I would very much like to see this section improved as it is important for the paper as a whole.

I also feel that some of the assumptions that have gone in to the evacuation modelling are rather optimistic, for example the assumption of no reaction time. The discussion section makes it clear that the authors are aware of many of these issues, but some further elaboration may be useful.

Thanks for these relevant comments. We have tried to improve our explanations and added some further thoughts on the issue of the evacuation time at the end of the discussion (Section 4.3, lines 470 – 473), as detailed below in our replies to your specific comments.

**Specific Comments**

Landslide modelling

A challenge with landslide modelling is the wide range of possible initial parameters. The paper considers variations in three landslide parameters: the position upslope, the volume, and the density. The authors mention that off-axis landslide position and rheology are parameters for future study.

However lines 333-334 could perhaps be revised to be more clear that the volume and position were found to be the most important parameters of those considered *in this study*. As it is not clear from this study alone that all other parameters are 'of second order'.

We clarified the text as "In this study, landslide position and volume are considered the key parameters in determining the hazard score (Cerminara et al., 2024)". This has been added in Section 4.1, lines 399 – 400, with adding the reference of the INGV database recently published.

The result that it is landslides from the lowest subaerial position (position 3) caused the highest impact is interesting and a bit surprising. The authors assert that this is related to the landslide not having time to deform before reaching the sea surface. This is a plausible explanation to me, although it would be nice to see this demonstrated with a figure or reference. Since the rate of deformation is related to

the rheology, it also makes me question whether the effect of rheology is truly 'second order' (see previous paragraph).

Following the comments made for Reviewer 1, who also raised this interesting point, this is now addressed by adding a short discussion (Section 4.1, lines 390 – 398) and two new appendices (Appendix D and E).

Analysis of Signals

The use of LASSO penalised linear regression to improve the prediction of inundation is interesting and could be a useful tool in many circumstances. The explanation of the method in the main text and the appendix was quite brief, and the supplied reference Giraud (2021) also rather hard to follow, so any additional explanation of how the method was applied to this problem would be welcome.

In a real case, the volume, density or position of the landslide is not immediately known, thus we only have the signals of the two gauges. As a result, the aim of LASSO is to find out whether there is any chance of detecting the impact of tsunamis on coastlines both quickly and accurately (without counterproductive false alarms) before the tsunami arrives and without knowledge of the landslide characteristics. We thus determine how we can use simulated wave signals to correctly determine the impact score. Consequently, our aim is not to analyse the shape of the waveforms as a function of the physical characteristics of the landslide.

To set the impact score, we used the signals from the 2 beacons, i.e., 40 variables (1 wave height every 2 seconds between 0 and 40 seconds for 2 beacons). This is now explained in the text.

The classic method for this type of problem (that is, to find an unknown value from several knowns) is linear regression. This approach, however, lacks robustness when the number of explanatory variables (here, 40) is too large compared with the number of individuals (here, the 156 simulations). LASSO linear regression is thus a modification of traditional linear regression that identifies a subset of explanatory variables (in this case, times of interest for measuring wave heights) of sufficiently small size for the results to be robust. This has been clarified through addition of text to the methods (Section 2.2, lines 133 – 143) and results sections (Section 3.3, lines 296 – 300), as also recommended by reviewer 1.

My main concern here is how robust the LASSO regression algorithm would be in scenarios that differ in one way or another from those on which it has been trained. For example: landslides that occur off axis, or have different rheologies to those assumed; or how the algorithm would work if there was not a singular landslide but one quickly followed by another (as in 2002 but with a shorter gap) such that there was a superposition of waves.

Our LASSO method is robust in terms of ability to adapt to landslides with volumes of between 5 and 30 million $m^3$, densities of between 1.7 and 2.5 kg/$m^3$ and landslide source positions of between 500 m and - 584 m. That is, within the limits of our simulations. This has been added in Section 3.3, lines 267 – 270.

As argued in a recent paper by Ripepe and Lacanna (2024) the waveform did not change with the slip volume and that it was possible to determine the inundation extent/run-up from the tsunami amplitude, which we also show here. See Section 3.3, lines 270 – 279, for more details.

Another question is how to determine time t=0 instrumentally from the water-level data, and hence when to extract data to use in the algorithm. There is also a potentially longer wait to collect all of the datapoints which may limit the use in some near-source cases. To study all of these things would require another paper, for now I just suggest the authors consider a bit more discussion.

Currently LASSO needs 40 seconds to recognize and classify the waveform. This time was a result of a payoff between precision in inundation area and time needed to classify the waveform to an acceptable degree of accuracy. We could reduce the time, but that would make output decreasingly accurate. This means that for extreme proximal locations output will be delivered after wave arrival, but they will have been alerted by the siren. Discussed now in Section 4.3, lines 459 – 462.

In Appendix A, I was confused for a while by the way that 'X' was used both for the inundation of cells and for the tsunami detector time series. Maybe use a different letter for the inundations and clarify how 'Y' is calculated from it?

X is a vector, and Y is calculated as a function of the various vectors. The impact score is first calculated from the inundation models, and then we find the impact score for the signal associated with each inundated pixel with vector X. Thus, X is the same for both the inundation of cells and the detector time series.

Evacuation Capacity

I found Section 3.4 quite hard to follow, mostly because the concept of 'warning time' (both 'real' and 'needed') was introduced without really clear definitions. With some effort I could establish what I think these are, but I think it would be better to spell this out more explicitly. Similarly it would be good to be fully clear about what the maximum and minimum of these times were calculated.

Defined and clarified in the beginning of Section 3.4, lines 308 – 314. We define "warning time needed" and "real warning time". The former is the time needed to move from any given point to a safe point, this being the exit from the inundation zone Refuge Area Entry Point (RAEP). Instead, the "real time", is the time available for escape prior to wave arrival. Also, instead of using "evacuable" or "non-evacuable" we use "inside threshold time" and "outside threshold time" to avoid appearing to make any statement regarding evacuation.

Based on my interpretation (which could be wrong) I found figure 11 a bit difficult to understand. As by my understanding, the cells that need the most warning time (hollow bars) are unrelated (or even inversely related) to those which need the most warning time (solid bars), yet because they appear next to each other in the figures I initially thought that they would be the same locations.

The figure is designed to portray the cases where we lack the time to evacuate a point as a function of the time of the day and/or zone. This is now clarified in the figure caption (Section 3.4, lines 359 – 360).

Pedestrian Evacuation Model

The evacuation model used is relatively simple, and the most important thing for this paper is to make sure that the simplifications and approximations are well documented and explained. Simplifications I'm aware of (some of which are mentioned in the text) include:

- Assumption of no reaction time

- Assumption of no warning dissemination time

- Assumption of no variation in walking speed

- Assumption of no congestion

While some of these could be approximated by extending the current model, ultimately this problem is really calling out for a full agent-based modelling approach (though I'm not suggesting that the authors need to do that for this paper).

We are currently testing such problems using on-site escape tests using multiple agents faced with differing traffic flow scenario's, starting positions (beach, water, bed), and degree of preparation (footware, reaction time). This will enable us to better calibrate our escape times, and to produce a distribution of potential times depending on traffic and route conditions. We now address this issue at end of the discussion (Section 4.3, lines 470 – 473) and as also following the advice of reviewer 1.

Summary

Although I have made some critical comments above, in all I find this to be a valuable and well-written multi-disciplinary paper, certainly worthy of publication after some minor adjustments.